# RL's Razor: Why Online Reinforcement Learning Forgets Less

**Idan Shenfeld**[*]    **Jyothish Pari**[*]    **Pulkit Agrawal**
Improbable AI Lab, MIT
{idanshen, jyop, pulkitag}@mit.edu

## Abstract

Comparison of fine-tuning models with reinforcement learning (RL) and supervised fine-tuning (SFT) reveals that, despite similar performance at a new task, RL preserves prior knowledge and capabilities significantly better. We find that the degree of forgetting is determined by the distributional shift, measured as the KL-divergence between the fine-tuned and base policy evaluated on the new task. Our analysis reveals that on-policy RL is implicitly biased towards KL-minimal solutions among the many that solve the new task, whereas SFT can converge to distributions arbitrarily far from the base model. We validate these findings through experiments with large language models and robotic foundation models and further provide theoretical justification for why on-policy RL updates lead to a smaller KL change. We term this principle *RL's Razor*: among all ways to solve a new task, RL prefers those closest in KL to the original model.

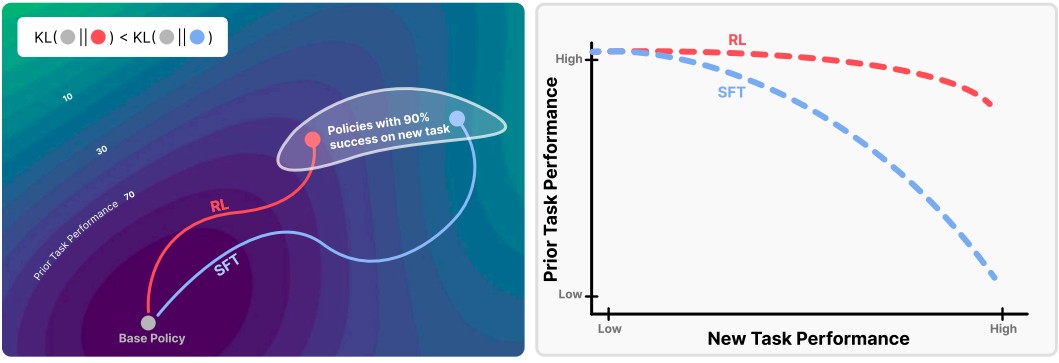

Figure 1: **Bias toward KL-minimal solutions reduces forgetting.** *Left:* Among policies that solve the new task, RL converges to those closest in KL to the base model. *Right:* This KL bias yields higher prior-task retention at matched new-task performance compared to SFT.

## 1 Introduction

Foundation models have rapidly become the backbone of modern AI, powering applications in language, vision, robotics, and beyond. Despite their remarkable capabilities, today's models are largely *static* once deployed: they excel at tasks learned during pre-training or post-training, but are not designed to self-improve and continually acquire new capabilities. We imagine a future where deployed models are long-lived *agents* assisting humans in the long-term and continuously adapting to new needs. As such, models must improve and adapt to new data, environments, and objectives Gao et al. (2025); Dao & Le (2025); Moradi et al. (2025); Li et al. (2025b); Simonds & Yoshiyama (2025); Zweiger et al. (2025).

---

[*]Equal contribution. Correspondence to: Idan Shenfeld (idanshen@mit.edu), Jyothish Pari (jyop@mit.edu).

A central challenge to this vision is *catastrophic forgetting*—the tendency for models to lose previously acquired capabilities when trained on new tasks McCloskey & Cohen (1989); French (1999); Kirkpatrick et al. (2017); Luo et al. (2023). Although scaling model size and pre-training data improves robustness Ramasesh et al. (2021); Luo et al. (2023); Cossu et al. (2024), catastrophic forgetting remains a persistent obstacle, undermining the promise of continual improvement Bommasani (2021); Guo et al. (2025b); Zweiger et al. (2025). To enable foundation models to serve as long-term agents, we need to develop post-training methods that allow models to acquire new skills without erasing old ones.

To further this goal, we analyze the performance of two widely used post-training schemes of supervised fine-tuning (SFT) and reinforcement learning (RL). Our experiments reveal a surprising finding: even when SFT and RL achieve the same performance on the new task, we observe that **SFT often achieves new-task gains by erasing prior knowledge, while RL better preserves old skills**. Figure 1 (right) illustrates this tradeoff: although both methods can reach high performance on the new task, RL maintains substantially higher performance on prior tasks compared to SFT.

This striking empirical gap raises the question: what underlying mechanism allows RL to improve on new tasks, but unlike SFT, minimally impacts the model's prior knowledge?

Previous approaches to catastrophic forgetting targeted specific factors such as constraining weight updates (Kirkpatrick et al., 2017; Aljundi et al., 2018; Zenke et al., 2017), preserving learned features (Rannen et al., 2017; Hou et al., 2019), or regularizing shift in output distribution (Li & Hoiem, 2017; Stiennon et al., 2020). While these methods can reduce forgetting, they focus on its effects rather than its underlying cause. Consequently, it remains unclear what truly governs forgetting or why different training algorithms behave so differently. Some prior work claimed that forgetting can be determined by how much the model's distribution shifts on past tasks (Rebuffi et al., 2017; Castro et al., 2018; Chaudhry et al., 2018; Wu et al., 2019). Yet in practice, this is infeasible to measure in foundation models, where the set of prior tasks is vast or even unbounded. To search for a more useful principle, we systematically ablated many candidate variables. Surprisingly, we find that forgetting can instead be predicted using only the *new* task distribution. Specifically, we uncover an *empirical forgetting law*: **When fine-tuning a model $\pi$ on a new task $\tau$, the degree of forgetting is accurately predicted by** $\mathbb{E}_{x \sim \tau}\left[\mathbf{KL}(\pi_0 \| \pi)\right]$, the KL divergence between the fine-tuned and base policy evaluated on the new task. This law is practically useful since it can be measured, and even influenced, during fine-tuning, without requiring access to past-task data. Although the mechanism remains to be fully understood, the consistency of this law across models and domains suggests it reflects a fundamental property of forgetting.

This law also clarifies the surprising difference between SFT and RL. Our analysis reveals a simple but powerful principle we call ***RL's Razor***: **among the many high-reward solutions for a new task, on-policy methods such as RL are inherently biased toward solutions that remain closer to the original policy in KL divergence**. Figure 1 (left) highlights this effect: among the many policies that reach a high success rate on the new task, RL is biased toward KL-minimal solutions, while SFT can converge to distant ones. This bias arises directly from RL's *on-policy training*: by sampling from the model's own distribution at every step, RL constrains learning to outputs already given non-negligible probability by the base model. To improve reward, these samples are reweighted and used to update the model, which gradually shifts the policy rather than pulling it toward an arbitrary distribution. Thus, when multiple equally good solutions exist for a new task, RL tends to find solutions close to the original policy, while SFT can converge to solutions much farther away, depending on the provided labels. Theoretical analysis in a simplified setting confirms this view, showing that policy gradient methods converge to KL-minimal solutions even without explicit regularization.

Finally, to validate the KL hypothesis, we construct an "oracle SFT" distribution that provably minimizes KL divergence while achieving perfect accuracy. Training on this oracle distribution produces even less forgetting than RL itself. This demonstrates that RL's advantage does not stem from being inherently different, but from its implicit KL minimization. Whenever training is biased toward KL-minimal solutions, forgetting is reduced.

Our main contributions are:

- We show that RL fine-tuning forgets less than SFT, even when both reach the same performance on new tasks.

- We uncover an empirical forgetting law: the KL divergence to the base policy, measured on the new task, as a strong predictor of catastrophic forgetting across objectives and hyperparameters.

- We provide empirical and theoretical evidence that the on-policy nature of policy gradient methods leads to smaller KL shifts and explains RL's advantage.

Together, these findings suggest a new perspective on post-training: to achieve continual adaptation without forgetting, algorithms should explicitly aim to minimize KL divergence from the base model. This principle opens the door to designing future training methods that combine RL's ability to preserve prior knowledge with the efficiency of SFT, enabling foundation models that can truly *learn for life*.

## 2 RELATED WORK

**Foundation Models and Post-training**   In modern deep learning, large-scale models pre-trained on broad, diverse datasets (usually termed Foundation models) serve as general-purpose backbones (Radford et al., 2021; Achiam et al., 2023; Touvron et al., 2023; Hu et al., 2023; Li et al., 2024a) with broad domain knowledge and some zero-shot learning abilities (Radford et al., 2018; Brown et al., 2020). However, pre-trained models may not directly meet the requirements of specific applications or align with domain-specific constraints. Post-training methods address this gap by adapting foundation models to downstream tasks through supervised fine-tuning on curated datasets (Howard & Ruder, 2018; Dodge et al., 2020; Wei et al., 2021; Chung et al., 2024), reinforcement learning from human or automated feedback (Ziegler et al., 2019; Ouyang et al., 2022; Guo et al., 2025a; Zhai et al., 2024), and other techniques (Rafailov et al., 2023). In this work, we study how different post-training methods affect forgetting, focusing on supervised fine-tuning and reinforcement learning.

**Catastrophic Forgetting.**   While fine-tuning primarily aims to improve performance on a new specific task, preserving the model's pre-existing general capabilities is equally critical. Unfortunately, fine-tuning often leads to catastrophic forgetting—a phenomenon where learning new information significantly deteriorates previously acquired knowledge McCloskey & Cohen (1989); French (1999); Kirkpatrick et al. (2017); Ouyang et al. (2022); Luo et al. (2023). Many works have sought to reduce forgetting by constraining updates, for example, by penalizing the magnitude of change in the model parameters, features, or matching the output on previous tasks/datasets (Wang et al., 2024). These methods are effective heuristics, but they address the symptoms of forgetting rather than explaining its cause. Our aim is to identify a simple and predictive metric that explains when and why forgetting occurs across different training algorithms.

We do not introduce a new training algorithm, but instead identify a simple *empirical forgetting law*: the KL divergence between the fine-tuned and base policy, measured *on the new task*, reliably predicts the degree of forgetting. The law also sheds light on why some mitigation strategies work. For example, methods like Elastic Weight Consolidation (Kirkpatrick et al., 2017) can be seen as approximations to KL minimization (Chaudhry et al., 2018). Interestingly, practitioners have also observed that KL regularization used in RL fine-tuning of LLMs as a heuristic for stabilizing optimization or preventing reward hacking Stiennon et al. (2020); Gao et al. (2023), also helps reduce catastrophic forgetting (Ouyang et al., 2022). Our contribution is to show that KL divergence is not merely a useful heuristic, but a reliable predictor of forgetting across settings.

**SFT versus RL.**   Prior comparisons between SFT and RL have focused on new task performance. A seminal result in sequential decision making is that on-policy learning can achieve stronger performance even when the expert providing supervision is the same one used to generate the offline dataset (Ross et al., 2011). Recent empirical studies have also found that RL fine-tuned models often exhibit superior generalization beyond the training distribution Han et al. (2025); Chu et al. (2025); Li et al. (2025a) and transfer more effectively to related tasks Huan et al. (2025) compared to SFT. However, prior works haven't examined the relative susceptibility of RL and SFT to catastrophic forgetting, which is the focus of our study.

Concurrently, Lai et al. (2025) reports that RL forgets less than SFT, but ascribes RL's advantage to learning from negative examples and not to the on-policy nature of RL. Results in Section 5

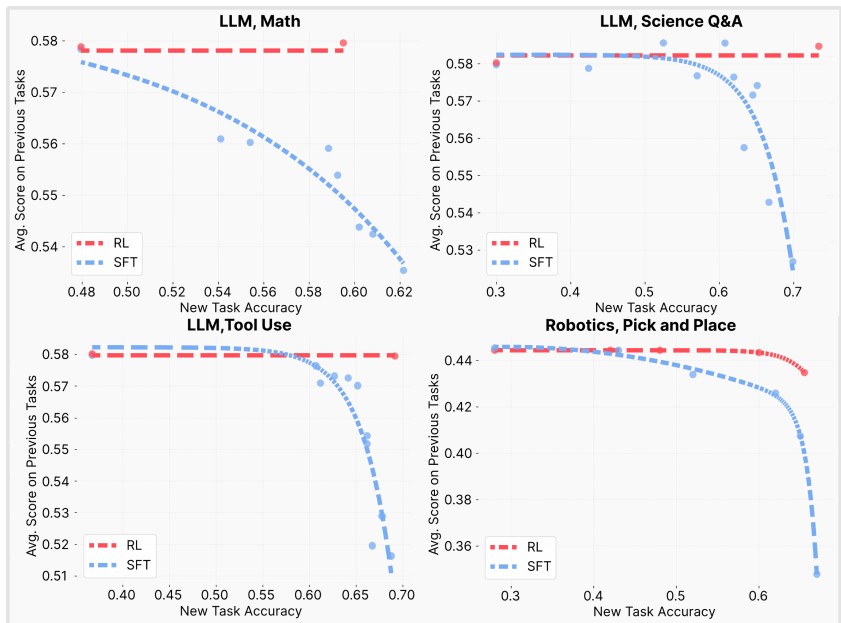

Figure 2: **Pareto frontiers of RL and SFT.** Comparing the performance of a fine-tuned model on the new task (x-axis) and prior task (y-axis). Each point corresponds to a model trained with a different set of hyperparameters, and the curves trace the Pareto frontiers for the two methods. RL achieves new-task improvements while maintaining prior knowledge, whereas SFT improves new-task performance at the expense of forgetting the prior task.

contradict their explanation of why RL forgets less, showing that the on-policy nature of RL is key. We also contribute the empirical forgetting law, the RL Razor, and its theoretical justification.

## 3 REINFORCEMENT LEARNING FORGETS LESS THAN SFT

We report results comparing the degree of catastrophic forgetting against new-task performance induced by RL and SFT on various large language model (LLM) and simulated robotic tasks.

### 3.1 PERFORMANCE TRADE-OFFS

**Experimental Setup.** For each new task, we fine-tuned models using the same set of prompts. One group of models was trained with SFT, and another with RL using GRPO Shao et al. (2024). Evaluation was performed along two axes:

- New task Performance: We measured performance on the held-out test set of the newly introduced task to assess the performance gain from the training.
- Previous tasks Performance: We measured performance on a diverse set of unrelated benchmarks. A drop in these benchmarks was taken as a measure of catastrophic forgetting.

Importantly, *our RL implementation uses GRPO with single gradient steps (effectively REINFORCE with normalized rewards), no KL penalty, and no clipping.*

Since different hyperparameters can lead to varying trade-offs between learning and forgetting, we trained dozens of models under diverse hyperparameter settings for both SFT and RL. To compare methods fairly, we identify the Pareto frontier in the two-dimensional plane of new-task performance versus previous-task performance. The Pareto frontier represents the set of models for which no further improvement on the new task is possible without incurring greater forgetting. Figure 2 (right) reports these frontiers: each point corresponds to a trained model with a different set of hyperparameters, and the Pareto-frontier curve indicates the best achievable trade-off for each method.

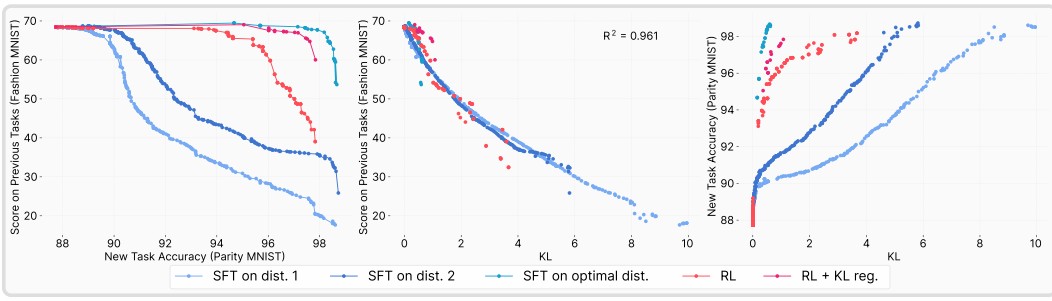

Figure 3: **KL divergence predicts catastrophic forgetting.** (Left) Learning-Forgetting Trade-offs. SFT outperform RL only when an oracle distribution is used as a source of annotation. (Middle) Forgetting aligns to a single curve when plotted against KL divergence, showing KL as a strong predictor across methods. (Right) RL improves new-task accuracy with much smaller KL shifts than SFT, highlighting the conservativeness of on-policy updates.

**Tasks and Datasets.** We perform experiments across three LLM and a single robotic tasks:

- *LLM, Math reasoning*: Qwen 2.5 3B-Instruct (Qwen et al., 2025) trained on math questions from the Open-Reasoner-Zero dataset (Hu et al., 2025).

- *LLM, Science Q&A*: Qwen 2.5 3B-Instruct trained on Chemistry L-3 subset of SciKnowEval (Feng et al., 2024).

- *LLM, Tool use*: Qwen 2.5 3B-Instruct trained on ToolAlpaca dataset (Tang et al., 2023).

- *Robotics, Pick and Place*: OpenVLA 7B (Kim et al., 2024) trained in the SimplerEnv environment (Li et al., 2024b) on the task of picking up a can.

To measure forgetting, we evaluated the finetuned models on established benchmarks covering diverse prior capabilities. For LLMs, we used Hellaswag (Zellers et al., 2019), TruthfulQA (Lin et al., 2021), MMLU (Hendrycks et al., 2020), IFEval (Zhou et al., 2023), Winogrande (Sakaguchi et al., 2021), and HumanEval (Chen et al., 2021). For robotic policies, we evaluated on the open/close drawer SimplerEnv tasks, excluding the one used for fine-tuning. These benchmarks act as proxies for prior skills that should be preserved during adaptation. Full details on SFT data sources, hyperparameters, and training/evaluation protocols are provided in Appendix C.

**Results.** Figure 2 reports the trade-off between new-task performance and retention of prior abilities. For RL, as accuracy on the new task increases, performance on previous benchmarks remains nearly unchanged. In contrast, SFT improvements on the new task consistently come at the cost of substantial forgetting. This difference is most pronounced in *Math*, where even small gains on the fine-tuned task correspond to a sharp reduction in prior-task performance. In *Science Q&A* and *Tool Use*, SFT retains some ability on prior tasks at lower accuracy levels for the new task, but performance deteriorates rapidly as the model approaches higher accuracy on the new task.

> **Takeaway 1**
>
> RL is able to learn new tasks while incurring minimal forgetting, whereas SFT reaches similar new-task performance only by sacrificing prior knowledge.

## 4 SMALLER KL DIVERGENCES LEAD TO LESS FORGETTING

As shown in Section 3, RL fine-tuning achieves comparable new-task performance to SFT while consistently forgetting less. Explaining this gap requires identifying a variable that determines the degree of forgetting across methods. We therefore searched for a predictor that could account for forgetting independently of the training algorithm or hyperparameters. Such a predictor would both explain the empirical difference between RL and SFT and offer a unifying principle for catastrophic

forgetting. Prior work has proposed candidates such as the magnitude of weight changes, sparsity of updates, or gradient rank. Across our experiments, however, none of these variables consistently aligned with the observed forgetting behavior (see Section 6). What did emerge was an *empirical forgetting law*: the **KL divergence between the fine-tuned model and the base model, measured on the new task**, reliably predicts the degree of forgetting.

It seems more natural to measure KL on prior tasks, and not on new tasks, and indeed that was the focus of multiple prior works (Li & Hoiem, 2017; Kirkpatrick et al., 2017). However, since foundation models undergo many training stages, old-task distributions are ill-defined or inaccessible. Moreover, we've found that prior task KL and new task KL are highly correlated, as can be seen in Figure 13.

Testing this hypothesis in large LLMs is challenging, since RL training is computationally expensive and cannot easily be run to convergence. Moreover, the search for predictors requires repeating fine-tuning many times under diverse conditions. To address these limitations, we designed a controlled toy setting, ParityMNIST, that allows us to replicate the RL–SFT gap under full convergence and perform systematic ablations.

ParityMNIST is derived from MNIST (Deng, 2012), but reframes the task as predicting parity (even vs. odd). An image of an even digit is correctly classified if the model predicts *any* even digit label, and likewise for odd digits. Multiple output distributions are thus equally valid, mirroring a key property of the generative tasks we studied in section 3: *many distinct policies can achieve the same performance*.

We pretrained a 3-layer MLP jointly on a subset of ParityMNIST and FashionMNIST (Xiao et al., 2017), then fine-tuned only on ParityMNIST while measuring forgetting on FashionMNIST. This design provides a minimal, tractable setting for investigating predictors of forgetting. To parallel the main experiments:

- In the **SFT** setting, the model was trained on labels sampled from a single arbitrary distribution out of the many possible correct ones.

- In the **RL** setting, the reward was correctness with respect to parity, leaving the model free to converge to any valid distribution.

For more details, see Appendix C.3. This design allowed us to replicate the phenomenon where RL reached high accuracy on the new task with substantially slower degradation of prior knowledge, while SFT exhibited a steeper trade-off (Figure 3, left). Importantly, *reproducing the effect in this simple MLP setting shows that it is not specific to large scale transformers, but a more general property of fine-tuning deep generative models*.

**KL as Predictor.** Plotting forgetting against the KL divergence from the base model on ParityM-NIST reveals a single functional relationship across both RL and SFT (Figure 3, middle). This indicates that forgetting is determined by KL divergence, not by the choice of training algorithm. A quadratic fit achieves $R^2 = 0.96$ in this setting, underscoring the strength of the relationship. To test robustness, we repeated the experiment with two different arbitrary SFT labelings. Although their Pareto frontiers differed, the forgetting–KL curves coincided, confirming that KL consistently predicts forgetting irrespective of training method or label distribution. The same correlation appears in our LLM experiments, with a quadratic fit achieving $R^2 = 0.71$ (Figure 12). While weaker, the residuals are mean-zero and can be attributed to noise from approximate KL and accuracy estimation.

**Optimal SFT Distribution.** To validate that KL divergence is the predictor variable, we constructed an oracle SFT distribution. In ParityMNIST, the simplicity of the task allows us to analytically identify the labeling that minimizes KL divergence to the base model among all distributions achieving 100% accuracy (Appendix C.3). If KL divergence fully determines forgetting, then training SFT on this oracle distribution should yield the optimal accuracy–forgetting trade-off. The results in Figure 3 confirm this prediction—SFT trained on the oracle distribution retained more prior knowledge than RL, achieving the best trade-off observed. RL performs well because its on-policy updates bias the solution toward low-KL regions, but when SFT is explicitly guided to the KL-minimal distribution, it can surpass RL. As an additional validation, we trained an SFT model

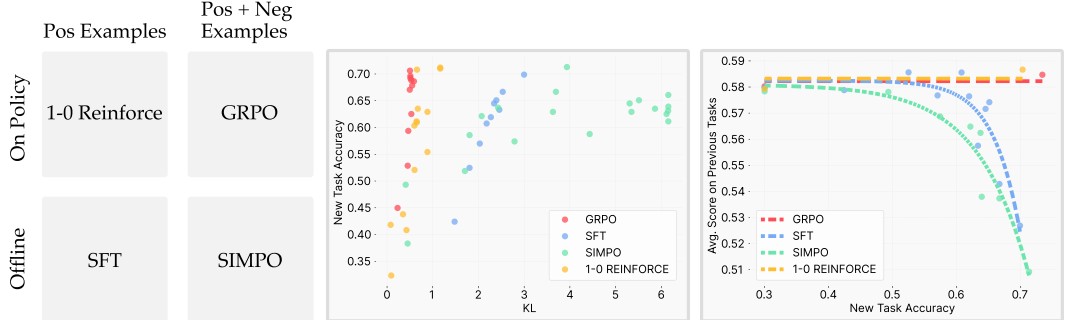

Figure 4: **Comparison of algorithm classes.** (Left) The four quadrants illustrate algorithm types, defined by whether they are on-policy or offline and whether they incorporate negative gradients. (Middle) On-policy methods retain prior knowledge more effectively. (Right) Both GRPO and 1-0 Reinforce achieve higher new-task accuracy while incurring smaller KL shifts from the base model, showing that on-policy methods consistently induce more conservative KL updates.

on data generated by an RL-trained model. The distilled SFT matched RL's accuracy–forgetting trade-off (Figure 10), reinforcing that the distribution learned, rather than the optimization algorithm, governs forgetting. Finally, we also added KL regularization to SFT, and found that it only minimally improves the forgetting-learning Pareto frontier (Appendix A).

> **Takeaway 2**
>
> Catastrophic forgetting in both SFT and RL is predicted by the KL divergence between the fine-tuned and base models on the new task.

## 5    ON-POLICY METHODS LEADS TO SMALLER KL DIVERGENCE

Having established that the KL divergence between the trained model and its base distribution on the new task predicts catastrophic forgetting, we now ask: why are RL fine-tuned models able to achieve strong task performance while moving less in KL than SFT models?

### 5.1    EXPERIMENTAL EVIDENCE

To understand the difference in KL behavior, it is useful to contrast the training objectives of SFT and RL. For discrete outputs, SFT minimizes cross-entropy against a supervision distribution $\pi_\beta$ over a distribution of inputs $\mathcal{D}$:

$$\mathcal{L}_{\text{SFT}}(\pi) = -\mathbb{E}_{x \sim \mathcal{D}, y \sim \pi_\beta}[\log \pi(y|x)]$$

In contrast, RL with policy gradients optimizes[*]:

$$\mathcal{L}_{\text{RL}}(\pi) = -\mathbb{E}_{x \sim \mathcal{D}, y \sim \pi}[A(x, y) \log \pi(y|x)]$$

where $A(x, y)$ is an Advantage function, which is the reward of $y$ normalized with respect to other rewards for the same $x$. Two features distinguish this from SFT:

1. Sampling Distribution. While in RL the training was done on outputs drawn from the model's own distribution, in SFT they come from fixed external annotations.

2. Negative Examples. While sampling from $\pi$, some of the responses will be incorrect. These are usually assigned a negative coefficient $A(x, y)$. This pushes probability mass away from poor outputs, a mechanism absent in SFT.

---

[*]Notice that in practice, the policy gradient trick (Sutton et al., 1998) ensures gradients are taken only through the log-probability term, not through the sampling distribution inside the expectation.

Our hypothesis is that one of these two differences is what causes RL's resistance to forgetting. To examine our hypothesis, we perform experiments with four different objectives:

- *GRPO*. An on-policy objective that utilizes negative examples. Here, $A(x, y)$ is the normalized reward.

- *1–0 Reinforce*. An on-policy algorithm that does not use negative examples. Here, $A(x, y) = 1$ for correct responses and $0$ for incorrect ones. This is equivalent to sampling from the model and performing SFT on correct answers only.

- *SFT*. An offline objective that does not use negative examples.

- *SimPO*. An offline objective that utilizes negative examples. We create negative examples by sampling incorrect responses from an external model, and use the SFT data for positive examples. The SimPO (Meng et al., 2024) loss compares correct and incorrect outputs via a logistic term:

$$\mathcal{L}_{\text{SIMPO}}(\pi) = -\mathbb{E}_{x \sim \mathcal{D}, y_w \sim \pi_{\beta+}, y_l \sim \pi_{\beta-}} \left[ \log \sigma \left( \log \pi(y_w|x) - \log \pi(y_l|x) - 1 \right) \right]$$

where $\pi_{\beta+}$ and $\pi_{\beta-}$ denote distributions for correct and incorrect responses, respectively. We used SimPO rather than naïve likelihood/negative likelihood because the latter was unstable to train.

We compared the four objectives on the Science Q&A task, measuring their learning–forgetting trade-offs as in Section 4. The results, shown in Figure 4, reveal that 1–0 Reinforce behaves similarly to GRPO, while SimPO resembles SFT. Thus, the critical factor is not the presence of negative gradients but the use of on-policy data. Plotting KL divergence confirms this conclusion: on-policy methods (GRPO and 1–0 Reinforce) reach the same task performance with significantly smaller KL divergence from the base model than offline methods (SFT and SimPO).

## 5.2 THEORETICAL PERSPECTIVE

Beyond the empirical results, it is useful to ask why on-policy methods naturally induce smaller KL shifts. One way to see this is through the lens of projection in probability space: policy gradient methods can be understood as a conservative projection that keeps the policy close to its starting point while reweighting toward higher-reward outcomes. At each step, the policy samples outputs it already finds likely, then re-weights those samples according to reward, shifting probability mass toward higher-reward outcomes while suppressing lower-reward ones. Crucially, because updates are defined relative to the model's own distribution, they nudge the policy toward a nearby re-weighted distribution, rather than pulling it toward a potentially distant external distribution (as in SFT). This explains why policy gradient methods tend to remain close to the base model in KL divergence.

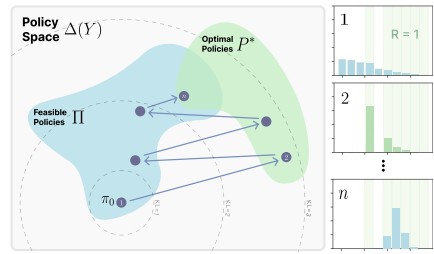

Figure 5: **KL-minimal path to optimality.** Alternating I-projection into the set of optimal policies and M-projection into $\Pi$ carries $\pi_0$ into $P^*$ while preferring the closest solution in KL.

This perspective can be formalized by observing that, in the binary-reward case, the re-weighted distribution targeted by policy gradient is exactly the minimum-KL projection of the current policy onto the set of optimal ones.

**Lemma 5.1.** *Let $p$ be a distribution over a finite set $Y$, and let $R : Y \to \{0, 1\}$ be a reward function. Rejection sampling from $p$ with acceptance condition $R(y) = 1$ yields a distribution $q_{\text{RS}}$. This distribution can be equivalently characterized as the solution to:*

$$q_{RS} = \arg\min_{q} D_{KL}(q||p) \quad s.t \quad \mathbb{E}_{y \sim q}[R(y)] = 1$$

Building on this, we show that policy gradient converges to the KL-minimal optimal policy within the representable family. A detailed version with proofs is provided in Appendix B.

**Theorem 5.2.** *Let $Y$ be a finite set and let $\Pi \subseteq \Delta(Y)$ be a convex family of feasible policies (e.g., an exponential family). Let $R : Y \to \{0, 1\}$ be a binary reward function and $P^* = \{q : \mathbb{E}_q[R] = 1\}$*

*the set of optimal policies. Then, under suitable regularity conditions, solving the reinforcement learning objective with policy gradient converges to*

$$\pi^\dagger = \arg \min_{\pi \in P^* \cap \Pi} D_{\mathrm{KL}}(\pi \,\|\, \pi_0),$$

*where $\pi_0$ is the initialization. In other words, policy gradient selects, among all optimal representable policies, the one closest in KL-divergence to the starting policy.*

Our theoretical analysis uses a simplified convex setting to isolate the on-policy effect. Extending our analysis to fully characterize deep network dynamics remains important future work.

> **Takeaway 3**
>
> On-policy training explains why RL maintains smaller KL divergence than SFT. Sampling from the model's own distribution keeps it close to the base model, while SFT pushes it toward arbitrary external distributions.

## 6 ALTERNATIVE HYPOTHESIS

Science advances not only by identifying the right explanations, but also by eliminating incorrect ones. To this end, we systematically evaluated alternative variables as potential predictors of catastrophic forgetting, grouped into four categories:

- **Weight-level changes.** Many prior work tried to mitigate forgetting by constraining the change in parameter space (Kirkpatrick et al., 2017; Aljundi et al., 2018; Zenke et al., 2017). We measured parameter changes under $L_1$, Fisher-weighted $L_2$, and spectral norm metrics. The Fisher matrix was computed on the basis of the model parameters, with expectation over inputs from the previous task. These metrics correlated only weakly with forgetting: large parameter shifts could occur without forgetting, and conversely, forgetting sometimes occurred despite small parameter movement.

- **Representation-level changes.** Some other papers focused on maintaining the previous features (Jung et al., 2018; Hou et al., 2019; Dhar et al., 2019). We examined hidden activation shifts (L1 and L2 distances) as proxies for changes in internal representations. Although we found that there is representation drift during training (see Appendix D.1), the curves were distinct between training objectives, meaning that it is not a good predictor.

- **Sparsity and rank of updates.** Motivated by Mukherjee et al. (2025), who argue that RL updates are sparse while SFT weight updates are dense, we explicitly tested this hypothesis. We found that the reason for the observed sparsity was the use of `bfloat16` for model training. Since `bfloat16` has a limited mantissa, small parameter updates (such as those produced by RL) can fail to cross the representational threshold, effectively causing no update at all. Performing the same training with `float32` resulted in models with identical performance but without any sparsity in their weight updates. The rank of all weight updates was full.

- **Distributional distances.** We considered multiple measures of output distribution change, all measured over inputs from the new task $\tau$: Forward KL ($\mathbb{E}_{x\sim\tau}\big[\mathrm{KL}(\pi_0\|\pi)\big]$), Reverse KL ($\mathbb{E}_{x\sim\tau}\big[\mathrm{KL}(\pi\|\pi_0)\big]$), Total Variation, and $L_2$ distance between distributions.

Table 1 summarizes these results for the MNIST task. Across all candidates, KL divergence (both forward and reverse) between the fine-tuned and base model evaluated on the new task emerges as the only consistent and high-fidelity predictor of catastrophic forgetting.

## 7 DISCUSSION AND CONCLUSION

Our study reveals that catastrophic forgetting is governed not by the choice of training algorithm, but by the KL divergence from the base policy evaluated on the new task. This explains why RL forgets less than SFT, as on-policy training naturally biases updates toward KL-minimal solutions, preserving prior knowledge while acquiring new skills.

However, we still lack a mechanistic account of why larger KL shifts on the new task disrupt prior knowledge—whether through representational interference, implicit capacity limits, or other dynamics. Moreover, while we demonstrate the KL–forgetting link across moderate-scale LLMs and

| Variable | $R^2$ (2nd deg. polynomial) |
|---|---|
| KL, forward | **0.96 ± 0.01** |
| KL, reverse | 0.93 ± 0.01 |
| TV | 0.80 ± 0.01 |
| Distribution change, L2 | 0.56 ± 0.02 |
| Weight change, L1 | 0.34 ± 0.02 |
| Weight change, Fisher Weighted L2 | 0.58 ± 0.02 |
| Weight change, spectral norm | 0.58 ± 0.02 |
| Sparsity of weight change | N/A |
| Rank of weight change | N/A |
| Activation change, L1 | 0.52 ± 0.02 |
| Activation change, L2 | 0.55 ± 0.02 |

Table 1: Predictive power of alternative variables compared to KL.

toy models, its behavior at frontier scales and in more diverse generative domains remains unknown. In addition, we didn't study online but off-policy algorithms, which are popular in RL. Addressing these gaps will be essential for grounding the principle and extending it to real-world deployment.

Taken together, our results motivate a new design axis for post-training research: algorithms should be judged not only by how well they optimize new tasks, but also by how conservatively they move in KL relative to the base model. Importantly, this does not mean offline data cannot help, but that continual learning requires updates to keep learning close to the KL-minimal path. Embracing this principle may allow us to build agents that not only learn new skills, but also truly learn for life.

## 8 Use of Language Models

The authors used large language models to polish and revise the writing of the manuscript. The models were not used to generate ideas, perform analysis, or produce original scientific content.

## Acknowledgment

We want to express our gratitude to Nitish Dashora, Seungwook Han, Moritz Reuss, Zhang-Wei Hong, Leshem Choshen, Ahmad Beirami, Mehul Damani, Akarsh Kumar, and members of the Improbable AI lab for the helpful discussion on the paper. We are grateful to MIT Supercloud and the Lincoln Laboratory Supercomputing Center for providing HPC resources. The research was supported in part by Hyundai Motor Company, Qualcomm Innovation Fellowship, Google, and Amazon. The research was sponsored by the Army Research Office and was accomplished under Grant Number W911NF-21-1-0328. The research was also sponsored by the Office of Naval Research and was accomplished under Grant Number N00014-22-1-2740. Research was also sponsored by the Department of the Air Force Artificial Intelligence Accelerator and was accomplished under Cooperative Agreement Number FA8750-19-2-1000. The views and conclusions contained in this document are those of the authors and should not be interpreted as representing the official policies, either expressed or implied, of the Department of the Air Force or the U.S. Government. The U.S. Government is authorized to reproduce and distribute reprints for Government purposes notwithstanding any copyright notation herein. The views and conclusions contained in this document are those of the authors and should not be interpreted as representing the official policies, either expressed or implied, of the Army Research Office, Naval Research Office, Air Force, or the U.S. Government.

## Author Contributions

**Jyothish Pari** co-developed the project and contributed in all aspects of experiments and writing.

**Idan Shenfeld** co-developed the project and contributed in all aspects of experiments and writing.

**Pulkit Agrawal** co-developed the project direction, advised IS and JP, and played a significant role in paper writing.

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

## A  THE EFFECT OF KL REGULARIZATION

In our main experiments, we did not employ explicit KL regularization. Nevertheless, our finding that forgetting is closely predicted by the KL divergence to the base model naturally raises the question: can directly regularizing KL divergence mitigate forgetting? This is especially relevant given that KL penalties are widely used in reinforcement learning fine-tuning of large language models (Stiennon et al., 2020; Ouyang et al., 2022; Vieillard et al., 2020).

**Empirical observations.** We revisited the ParityM-NIST setup from Section 4, this time adding explicit KL penalties to both SFT and RL training. For each method, we conducted a hyperparameter sweep and varied the regularization coefficient over $0.1, 0.2, 0.5$. Figure A reports the resulting Pareto frontiers of the learning–forgetting trade-off. The effect is strikingly asymmetric:

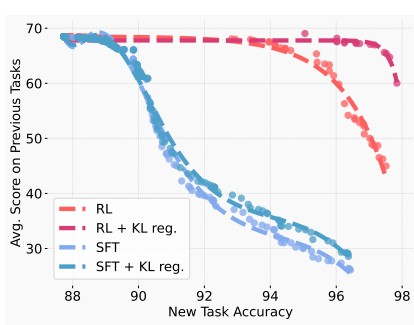

In RL, KL regularization substantially improves the trade-off. By explicitly discouraging large deviations from the base model, it amplifies RL's inherent bias toward KL-minimal solutions, enabling gains on the new task while preserving performance on prior tasks.

In SFT, KL regularization has only marginal effect. While it slightly restrains the model from drifting too far, the optimization remains tied to external supervision distributions, which may themselves be far from the KL-minimal solution. As a result, the overall frontier is essentially unchanged.

Figure 6: explicit KL regularization helps RL retain prior skills, but barely affects SFT training.

These results suggest that explicit KL regularization cannot rescue SFT from its fundamental limitation: SFT is forced to imitate whatever distribution is provided, and cannot search for new solutions.

**Theory**  This intuition can be formalized. RL with KL regularization effectively restricts optimization to policies achieving a given reward level and then selects the KL-minimal one. Thus, whenever the optimal reward is attainable, RL with a sufficiently small KL penalty converges to the minimum-KL optimal policy. By contrast, SFT with KL regularization minimizes cross-entropy to a fixed annotator distribution plus a KL penalty, and in general cannot guarantee alignment with the minimum-KL solution. Formally:

**Theorem A.1.** *Let $\Delta$ be the set of probability measures on $\mathcal{Y}$, and $\Pi \subseteq \Delta$ a nonempty feasible policy class. Fix a base policy $\pi_0 \in \Pi$ and a reward $R : \mathcal{Y} \to \mathbb{R}$. let*

$$R_{\max} = \sup_{\pi \in \Pi} \mathbb{E}_\pi[R], \qquad P^* = \{\pi \in \Pi : \mathbb{E}_\pi[R] = R_{\max}\}$$

*For $\beta > 0$ consider the RL with KL regularization objective:*

$$\pi_\beta^{RL} = \arg\max_{\pi \in \Pi} \mathbb{E}_\pi[R] \; - \; \beta \, \mathrm{KL}(\pi \| \pi_0)$$

*if $R_{\max}$ is attainable by the policy class then there exists $\bar{\beta} > 0$ such that for all $\beta \leq \bar{\beta}$,*

$$\pi_\beta^{RL} \in \arg\min_{\pi \in P^*} \mathrm{KL}(\pi \| \pi_0)$$

*Now, define annotator distribution $q \in \Delta$. For $\beta > 0$, consider the SFT with KL regularization objective:*

$$\pi_\beta^{SFT} = \arg\min_{\pi \in \Pi} -\mathbb{E}_{y \sim q}[\log \pi(y)] \; + \; \beta \, \mathrm{KL}(\pi \| \pi_0)$$

*In general, there is no $\beta > 0$ for which $\pi_\beta^{SFT}$ equals the minimum-KL optimal policy, even when $q$ itself is optimal ($\mathbb{E}_q[R] = R_{\max}$).*

# B  THEORY

## B.1  IMPLICIT BIAS OF ON-POLICY RL

**Lemma B.1** (Rejection sampling as an I-projection). *Let $p$ be a distribution over a finite set $Y$, and let $R : Y \to \{0, 1\}$ be a reward function. Rejection sampling from $p$ with acceptance condition $R(y) = 1$ yields a distribution $q_{RS}$. This distribution can be equivalently characterized as the solution to:*

$$q_{RS} = \arg\min_q D_{KL}(q||p) \quad s.t \quad \mathbb{E}_{y \sim q}[R(y)] = 1$$

*Equivalently, $q_{RS}$ is the I-projection of $p$ onto the set $\{q : \mathbb{E}_q[R] = 1\}$*

*Proof.* Let $S = \{y \in Y : R(y) = 1\}$. Rejection sampling produces the conditional distribution

$$q_{RS}(y) = \begin{cases} \frac{p(y)}{p(S)} & y \in S, \\ 0 & y \notin S, \end{cases}$$

where $p(S) = \sum_{y \in S} p(y)$ and we assume $P(S) > 0$.

Now consider the optimization problem. The constraint $\mathbb{E}_q[R] = 1$ means

$$\sum_{y \in Y} q(y)R(y) = \sum_{y \in S} q(y) = 1$$

so $q$ must put all of its mass on $S$. Thus the feasible set is exactly all distributions supported on $S$.

For any $q$ supported on $S$, we can write $p(y) = p(S)\,p(y|S)$ for $y \in S$, and then

$$D_{\mathrm{KL}}(q\|p) = \sum_{y \in S} q(y) \log \frac{q(y)}{p(y)} = \sum_{y \in S} q(y) \log \frac{q(y)}{p(y \mid S)} - \log p(S) \sum_{y \in S} q(y)$$

$$= D_{\mathrm{KL}}\big(q\|p(\cdot \mid S)\big) - \log p(S)$$

where we used $\sum_{y \in S} q(y) = 1$ in the last step. The second term is constant in $q$, so minimizing $D_{\mathrm{KL}}(q\|p)$ is the same as minimizing $D_{\mathrm{KL}}(q\|p(\cdot|S))$. By strict convexity of $D_{\mathrm{KL}}(\cdot\|\cdot)$ in its first argument, the unique minimizer is $q = p(\cdot \mid S) = q_{\mathrm{RS}}$. $\square$

**Lemma B.2** (Policy gradient as an M-projection). *Let $Y$ be a finite set and let $\Pi \subseteq \Delta(Y)$ be a set of admissible policies (distributions over $Y$). Consider the single-step reinforcement learning objective*

$$\max_\pi \mathbb{E}_{y \sim \pi}[R(y)]$$

*where $R : Y \to \mathbb{R}_{\geq 0}$ is a reward function. By the policy gradient theorem, this objective is equivalently optimized by*

$$\max_\pi \mathbb{E}_{y \sim \bar{\pi}}\big[R(y) \log \pi(y)\big]$$

*where $\bar{\pi}$ indicates that gradients are not propagated through the sampling distribution. Define the distribution*

$$q(y) = \frac{\pi(y)R(y)}{Z}, \qquad Z = \sum_{y \in Y} \pi(y)R(y)$$

*Then taking a policy gradient step is equivalent to taking a gradient step on the following objective:*

$$\min_\pi -\mathbb{E}_{y \sim q}[\log \pi(y)]$$

*In other words, as long a $q$ is fixed, optimizing the RL objective using policy gradient is equivalent to finding the $M$-projection of $q$ onto the set of feasible policies $\pi$ using gradient descent.*

*Proof.* Expanding the policy gradient objective gives

$$\mathbb{E}_{y \sim \bar{\pi}}[R(y) \log \pi(y)] = \sum_{y \in Y} \pi(y)R(y) \log \pi(y)$$

Let $Z = \sum_{y \in Y} \pi(y) R(y)$. Define $q(y) = \pi(y) R(y)/Z$. Then the above becomes

$$\sum_{y \in Y} \pi(y) R(y) \log \pi(y) = Z \sum_{y \in Y} q(y) \log \pi(y) = Z \, \mathbb{E}_{y \sim q}[\log \pi(y)]$$

Since $Z$ does not depend on $\pi$ in the gradient computation (it is treated as a constant in the $\bar{\pi}$ sense), maximizing the original objective is equivalent to maximizing $\mathbb{E}_{y \sim q}[\log \pi(y)]$.

Finally, recall that the $M$-projection of a distribution $q$ onto a set of distributions $\Pi$ is given by

$$\min_{\pi \in \Pi} \mathrm{KL}(q \| \pi) = \mathbb{E}_q[\log \frac{q}{\pi}] \; = \mathbb{E}_q[\log q] \; - \; \mathbb{E}_q[\log \pi]$$

since $\mathbb{E}_q[\log q]$ does not depend on $\pi$, the maximizer of $\mathbb{E}_{\bar{\pi}}[R \log \pi]$ over $\Pi$ coincides with $\arg\min_{\pi \in \Pi} \mathrm{KL}(q \| \pi)$. Thus, the policy gradient update corresponds to the $M$-projection of $q$ onto the policy class. $\qquad\square$

**Theorem B.3** (RL with binary reward as an EM algorithm). *Let $Y$ be a finite set and let $\Pi \subseteq \Delta(Y)$ be a set of feasible policies. Let $R : Y \to \{0, 1\}$ be a binary reward function and $P^*$ the set of all optimal policies $P^* = \{q : \mathbb{E}_q[R] = 1\}$. Then, solving the Single-step reinforcement learning objective using policy gradients is equivalent to performing the following optimization procedure:*

$$q_t = \arg\min_{q \in P^*} \mathrm{KL}(q \| \pi_t), \qquad \pi_{t+1} = \arg\min_{\pi \in \Pi} \mathrm{KL}(q_t \| \pi)$$

*This procedure is also known as EM with information projection.*

*Proof.* Sampling $y \sim \pi$ and accepting iff $R(y) = 1$ is exactly rejection sampling onto the event $S = \{y \in Y : R(y) = 1\}$. The resulting distribution is $\pi(\cdot | S)$. By Lemma A.1 with $p \leftarrow \pi$, this $\pi(\cdot | S)$ solves

$$\min_q D_{\mathrm{KL}}(q \| \pi) \quad \text{s.t.} \quad \mathbb{E}_q[R] = 1$$

establishing the I-projection. Applying Lemma A.2 on the RL objective gives us the M-projection. $\qquad\square$

**Proposition B.4** (Convergence to minimum KL solution). *Under the setting appear in theorem B.3 and assume $\Pi$ is an e-flat (exponential-family) model with full support, the optimal set $P^*$ is nonempty and realizable (i.e., $\Pi \cap P^* \neq \varnothing$). Then:*

*(1) If the M-projection is exact at every step, then $(\pi_t)$ converges to*

$$\pi^\dagger = \arg\min_{\pi \in P^* \cap \Pi} D_{\mathrm{KL}}(\pi \,\|\, \pi_0)$$

*(2) If the M-projection is inexact but, for some errors $\varepsilon_t \geq 0$, it holds that*

$$D_{\mathrm{KL}}(q_t \| \pi_{t+1}) \; \leq \; \min_{\pi \in \Pi} D_{\mathrm{KL}}(q_t \| \pi) \; + \; \varepsilon_t \quad \text{with} \quad \sum_{t=0}^{\infty} \varepsilon_t < \infty$$

*then $\pi_t$ also converges to the same limit $\pi^\dagger$.*

*Proof.* The I-step is always an exact I-projection (Lemma A.1). In the case of an exact M-step, the iterative process is EM with information projections. The e-/m-flat geometry yields the Pythagorean identities implying convergence to $\pi^\dagger$ (Dempster et al., 1977; Csiszár, 1984; Amari & Nagaoka, 2000). When the M-step only ensures a (near-)minimization up to summable errors, the iteration is GEM: monotone improvement and convergence follow from the GEM theory of Wu (1983) together with generalized alternating minimization for Bregman divergences (Gunawardana et al., 2005), which, under the same e-/m-flat assumptions, selects the same minimum-KL limit $\pi^\dagger$. $\qquad\square$

**Practical considerations.** Our theoretical equivalence should be interpreted with the following caveats:

- Beyond REINFORCE. In practice, many policy gradient algorithms such as GRPO and PPO replace the raw reward $R(y)$ with an advantage estimate $A(y)$. Since this substitution is a control variate technique, it leaves the expected gradient direction unchanged while reducing its variance. Thus, our projection-based interpretation continues to hold.

- In practice, the M-projection step is rarely done until convergence. More common is to collect a small dataset from the current policy, and then take a few gradient steps on mini-batches from this data before recollecting more.

- The optimal policy set $P^*$ defined by the linear constraint $\mathbb{E}_q[R] = 1$ is an $m$-flat family, but the representable policy set $\Pi$ induced by a neural network parametrization is not in general $e$-flat. This may prevent exact convergence to the minimum-KL solution described above. Nevertheless, our theorem provides a principled explanation for the bias observed in practical RL algorithms.

### B.2 KL REGULARIZATION

We will start by analyzing the setting of RL with KL regularization:

**Theorem B.5** (The solution to RL with KL regularization). *Let $\Delta$ be the set of probability measures on $\mathcal{Y}$, and $\Pi \subseteq \Delta$ a nonempty feasible policy class. Fix a base policy $\pi_0 \in \Pi$ and a reward $R : \mathcal{Y} \to \mathbb{R}$. For $\beta > 0$, consider the penalized problem*

$$\max_{\pi \in \Pi} \ \mathbb{E}_\pi[R] \ - \ \beta \operatorname{KL}(\pi \| \pi_0), \tag{1}$$

*and let $\pi_\beta^\star$ be any maximizer with value $\eta_\beta = \mathbb{E}_{\pi_\beta^\star}[R]$. Then $\pi_\beta^\star$ also solves the constrained problem*

$$\min_{\pi \in \Pi} \ \operatorname{KL}(\pi \| \pi_0) \quad s.t. \quad \mathbb{E}_\pi[R] = \eta_\beta. \tag{2}$$

*Conversely, if $\hat{\pi}$ solves equation 2 for some feasible target $\eta$, then there exists $\beta > 0$ such that $\hat{\pi}$ solves equation 1 and $\mathbb{E}_{\hat{\pi}}[R] = \eta$.*

*Proof.* From Korbak et al. (2022) we know that if $\Pi = \Delta$, the solution to Equation 1 is the exponentially tilted distribution

$$q_\beta(y) = \frac{\pi_0(y) e^{R(y)/\beta}}{Z_\beta} \qquad Z_\beta := \int e^{R(y)/\beta} \pi_0(y)$$

For the more general case where $\Pi \in \Delta$ whenever $Z_\beta < \infty$, we can write for any $\pi \in \Pi$:

$$\mathbb{E}_\pi[R] - \beta \operatorname{KL}(\pi \| \pi_0) = \int R\,\pi - \beta \int \log\left(\frac{\pi}{\pi_0}\right) \pi$$
$$= -\beta \int \log\left(\frac{\pi}{e^{R/\beta}\pi_0}\right) \pi$$
$$= -\beta \left(\operatorname{KL}(\pi \| q_\beta) - \log Z_\beta\right)$$

Hence maximizing equation 1 over $\Pi$ is equivalent to

$$\pi_\beta^* = \min_{\pi \in \Pi} \ \operatorname{KL}(\pi \| q_\beta).$$

By optimality, for every $\pi \in \Pi$,

$$\operatorname{KL}(\pi_\beta^\star \| q_\beta) \ \leq \ \operatorname{KL}(\pi \| q_\beta).$$

Using the decomposition $\operatorname{KL}(\pi \| q_\beta) = \operatorname{KL}(\pi \| \pi_0) - \beta^{-1} \mathbb{E}_\pi[R] + \log Z_\beta$, we obtain for all $\pi \in \Pi$:

$$\operatorname{KL}(\pi_\beta^\star \| \pi_0) - \tfrac{1}{\beta} \mathbb{E}_{\pi_\beta^\star}[R] \ \leq \ \operatorname{KL}(\pi \| \pi_0) - \tfrac{1}{\beta} \mathbb{E}_\pi[R].$$

Rearranging,

$$\operatorname{KL}(\pi_\beta^\star \| \pi_0) \ \leq \ \operatorname{KL}(\pi \| \pi_0) - \tfrac{1}{\beta}\big(\mathbb{E}_\pi[R] - \eta_\beta\big) \qquad \forall \pi \in \Pi$$

Now, fix any $\pi \in \Pi$ such that $\mathbb{E}_\pi[R] = \eta_\beta$. Plugging this equality into the inequality above kills the last term and yields

$$\mathrm{KL}(\pi_\beta^\star \| \pi_0) \leq \mathrm{KL}(\pi \| \pi_0)$$

i.e., among all $\pi \in \Pi$ with $\mathbb{E}_\pi[R] = \eta_\beta$, $\pi_\beta^\star$ minimizes $\mathrm{KL}(\cdot \| \pi_0)$. This proves that $\pi_\beta^\star$ solves equation 2.

Now for the other direction. Suppose $\hat{\pi} \in \Pi$ solves equation 2 at some feasible $\eta$. Consider the Lagrangian:

$$\mathcal{L}_\lambda(\pi) = \mathrm{KL}(\pi \| \pi_0) + \lambda(\eta - \mathbb{E}_\pi[R])$$

By the KKT conditions for the equality constraint, there exists $\lambda^\star > 0$ such that $\hat{\pi}$ minimizes $\mathcal{L}_{\lambda^\star}$ over $\Pi$. Equivalently, $\hat{\pi}$ maximizes $\mathbb{E}_\pi[R] - \beta \mathrm{KL}(\pi \| \pi_0)$ over $\Pi$ with $\beta = 1/\lambda^\star$, and necessarily $\mathbb{E}_{\hat{\pi}}[R] = \eta$ holds at the maximizer. Thus $\hat{\pi}$ solves equation 1. $\qquad\square$

**Corollary B.6.** *Assume there exists $\pi^* \in \Pi$ achieving the maximal attainable expected reward $R_{\max} := \sup_{\pi \in \Pi} \mathbb{E}_\pi[R]$. Let $\Pi_{\mathrm{opt}} := \{\pi \in \Pi : \mathbb{E}_\pi[R] = R_{\max}\}$. Then there exists $\bar{\beta} > 0$ such that for every $\beta \in (0, \bar{\beta}]$, any maximizer $\pi_\beta^\star$ of equation 1 satisfies $\mathbb{E}_{\pi_\beta^\star}[R] = R_{\max}$ and*

$$\pi_\beta^\star \in \arg\min_{\pi \in \Pi_{\mathrm{opt}}} \mathrm{KL}(\pi \| \pi_0).$$

*In words: once the KL penalty is small enough that the optimal reward is still achievable, the KL-regularized objective selects the* minimum-KL *optimal policy.*

Now we will move to analyzing the setting of SFT with KL regularization:

**Lemma B.7** (The solution to SFT with KL regularization). *Let $\Delta$ be the set of probability measures on $\mathcal{Y}$, and $\Pi \subseteq \Delta$ a nonempty feasible policy class. Fix a base policy $\pi_0 \in \Pi$ and $q$ as the distribution producing the annotations for the SFT training. For $\beta > 0$, consider the following objective:*

$$\min_{\pi \in \Delta(\mathcal{Y})} \quad -\mathbb{E}_{y \sim q}[\log \pi(y)] \; + \; \beta \, \mathrm{KL}(\pi \| \pi_0) \tag{3}$$

*Assume $\Pi = \Delta$, then the unique minimizer $\pi_\beta^\star$ is given by*

$$\pi_\beta^\star(y) \; = \; \frac{q(y)}{\beta \, W\big(\frac{q(y)}{\beta \, A(\beta) \, \pi_0(y)}\big)} \tag{4}$$

*where $W$ is the principal branch of the Lambert $W$ function and the scalar $A(\beta) > 0$ is chosen to satisfy the normalization $\sum_y \pi_\beta^\star(y) = 1$.*

*If $q(y) = 0$ for some $y \in \mathrm{supp}(\pi_0)$, the formula holds in the limit, yielding $\pi_\beta^\star(y) \to \pi_0(y)$. If $\pi_0(y) = 0$ and $q(y) > 0$, the objective value is $+\infty$ and no finite minimizer exists.*

*Proof.* Write the Lagrangian

$$\mathcal{L}(\pi, \lambda) = -\sum_y q(y) \log \pi(y) + \beta \sum_y \pi(y) \log \frac{\pi(y)}{\pi_0(y)} + \lambda \left( \sum_y \pi(y) - 1 \right).$$

Stationarity $\partial \mathcal{L} / \partial \pi(y) = 0$ gives

$$-\frac{q(y)}{\pi(y)} + \beta \left( \log \frac{\pi(y)}{\pi_0(y)} + 1 \right) + \lambda = 0 \iff \log \frac{\pi(y)}{\pi_0(y)} = \frac{q(y)}{\beta \, \pi(y)} - \left( 1 + \frac{\lambda}{\beta} \right).$$

Let $A := \exp(1 + \lambda/\beta) > 0$. Then:

$$\pi(y) = \pi_0(y) A^{-1} \exp\big(\tfrac{q(y)}{\beta \, \pi(y)}\big) \to \pi(y) \exp\big(-\tfrac{q(y)}{\beta \, \pi(y)}\big) = \pi_0(y)/A$$

Setting $u_y := \frac{q(y)}{\beta \, \pi(y)}$ yields $u_y e^{u_y} = \frac{q(y)}{\beta A \, \pi_0(y)}$, whence $u_y = W\big(\frac{q(y)}{\beta A \, \pi_0(y)}\big)$ and thus equation 4.

The normalizer $A(\beta)$ is uniquely determined by $\sum_y \pi_\beta^\star(y) = 1$. Strict convexity of equation 3 on the simplex implies uniqueness. $\qquad\square$

**Theorem B.8** (SFT with KL reg. does not guarantee the minimum-KL optimal policy). *Let $R : \mathcal{Y} \to \mathbb{R}$ be a reward and $R_{\max} = \sup_y R(y)$. Define the minimum-KL optimal policy as the solution of*

$$\min_{\pi \in \Pi} \mathrm{KL}(\pi \| \pi_0) \quad s.t. \quad \mathbb{E}_\pi[R] = R_{\max} . \tag{5}$$

*Let $\pi_\beta^\star$ denote the unique minimizer of equation 3 from Lemma B.7. Then, in general, there is* no *$\beta > 0$ such that $\pi_\beta^\star$ equals the minimum-KL optimal policy.*

*Proof.* (*Counterexample*). Let $\mathcal{Y} = \{1, 2, 3\}$, $R(1) = R(2) = 1$, $R(3) = 0$. Take $\pi_0 = (0.6, 0.2, 0.2)$, hence $\pi^{\min \mathrm{KL}} = (0.75, 0.25, 0)$. Let $q = (0.5, 0.5, 0)$, which is optimal ($q(S) = 1$). We show that no $\beta > 0$ yields $\pi_\beta^\star = \pi^{\min \mathrm{KL}}$.

Using the stationary condition from the proof of Lemma B.7, and assume $\pi_\beta^\star = \pi^{\min KL}$. We get this set of equations:

$$-\frac{0.5}{0.75} + \beta \left( \log \frac{0.75}{0.6} + 1 \right) + \lambda = 0, \quad -\frac{0.5}{0.25} + \beta \left( \log \frac{0.25}{0.2} + 1 \right) + \lambda = 0$$

which has no solution, thus a contradiction. $\qquad\square$

*Counterexample.* Take $\mathcal{Y} = \{1, 2, 3\}$ with $R(1) = R(2) = 1$, $R(3) = 0$. Let $\pi_0 = (0.6, 0.2, 0.2)$, then the minimum-KL optimal policy is $\pi^{\min \mathrm{KL}} = (0.75, 0.25, 0)$. Choose an *optimal* annotator $q = (0.5, 0.5, 0)$.

Assume for contradiction that there exist $\beta > 0$ and $\lambda \in \mathbb{R}$ such that $\pi_\beta^\star = \pi^{\min \mathrm{KL}}$. The first-order condition from Lemma B.7 reads, for each $y$,

$$-\frac{q(y)}{\pi(y)} + \beta \left( \log \frac{\pi(y)}{\pi_0(y)} + 1 \right) + \lambda = 0.$$

Plugging $\pi^{\min \mathrm{KL}}(1) = 0.75$, $\pi^{\min \mathrm{KL}}(2) = 0.25$, $\pi_0(1) = 0.6$, $\pi_0(2) = 0.2$, $q(1) = q(2) = 0.5$, we get

$$-\frac{0.5}{0.75} + \beta \left( \log \frac{0.75}{0.6} + 1 \right) + \lambda = 0,$$

$$-\frac{0.5}{0.25} + \beta \left( \log \frac{0.25}{0.2} + 1 \right) + \lambda = 0.$$

Subtract the first equation from the second to eliminate $\lambda$. Using $\log \frac{0.75}{0.6} = \log \frac{0.25}{0.2} = \log(1.25)$, the $\beta$-terms cancel and we obtain

$$-2 - \left( -\tfrac{2}{3} \right) = -\tfrac{4}{3} \neq 0,$$

a contradiction. Hence no $\beta > 0$ yields $\pi_\beta^\star = \pi^{\min \mathrm{KL}}$ in this setting. $\qquad\square$

## C  TRAINING AND EVALUATION DETAILS

### C.1  LLM EXPERIMENTS

Unless otherwise stated, all reinforcement learning experiments were conducted using GRPO (Shao et al., 2024).

For the *Math* reasoning task, the training set provided final answers but lacked reasoning chains required for SFT training. To obtain these, we queried DeepSeek R1 (Guo et al., 2025a), sampling up to 16 responses per prompt and retaining a single response that matched the correct final answer. This yielded valid annotations for 96% of the dataset. For the *Science Q&A* task, we applied the same procedure with GPT-4o, obtaining correct annotations for the entire dataset.

To construct the learning–forgetting trade-off curves (e.g., Figure 2), we followed the protocol below:

1. Hyperparameter sweep. We trained multiple models under a broad sweep of hyperparameters (see Table 2).

2. New-task evaluation. For *Math* and *Science Q&A*, accuracy was measured by comparing the model's final answer to the ground truth, ignoring intermediate reasoning chains. For Tool Use, we extracted API calls from the output and matched them against ground-truth calls via regular expressions.

3. Previous-task evaluation. We assessed performance on unrelated benchmarks as described in Section 3.1, using the `Language Model Evaluation Harness` (Gao et al., 2024).

4. Pareto filtering. From the trained models, we retained only those lying within 2 accuracy points of the Pareto frontier.

5. Curve fitting. An exponential function was fit to the filtered points to produce the trade-off curves.

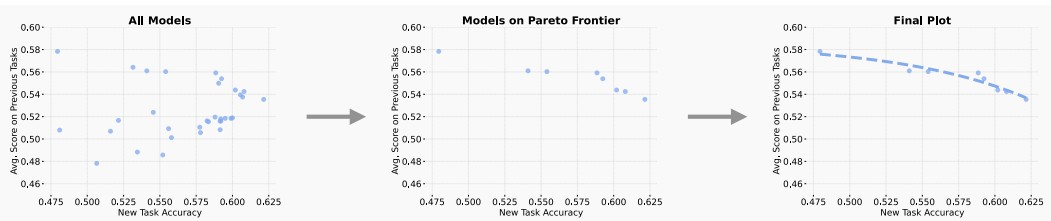

Figure 7: Example for the process of creating the pareto frontier plots

| Hyperparameter | SFT / SIMPO | RL |
|---|---|---|
| Base Model | Qwen2.5 3B-Instruct | Qwen2.5 3B-Instruct |
| Learning Rate | {1e-5, 3e-5, 5e-5, 7e-5, 9e-5} | {1e-5, 2e-5, 3e-5, 4e-5, 5e-5} |
| Optimizer | adamw | adamw |
| LR Scheduler | {constant w. warmup, cosine w. warmup} | constant w. warmup |
| Warmup steps | 50 | 50 |
| Epochs | {1,2} | 1 |
| Batch Size | {16,32,64,128} | See Below |
| Max Grad Norm | 1 | 1 |
| bfloat16 | True | True |
| Weight Decay | 0 | 0 |
| *GRPO-only hyperparameters* | | |
| KL reg. | | 0 |
| Group Size | | 64 |
| Prompts per generation | | 8 |
| num iterations ($\mu$) | | {1,2} |
| Loss type | | Dr. GRPO (Liu et al., 2025) |

Table 2: Hyperparameters used for the LLM experiments. Curly braces {} indicate a sweep over the specified values. Additional parameters such as weight decay and max gradient norm were manually ablated; since they showed no significant effect on results, they were not included in the final sweep.]

## C.2 ROBOTIC EXPERIMENTS

We evaluated the RL–SFT forgetting gap in a robotic control setting using the OpenVLA-7B model (Kim et al., 2024) as our base policy in the SimplerEnv environment (Li et al., 2024b). The fine-tuning task was a pick-and-place scenario requiring the robot to grasp and lift a can, while forgetting was measured on a distinct manipulation task of drawer opening/closing. This setting complements our LLM results by probing whether the KL–forgetting relationship generalizes to embodied policies. To construct the pareto-frontier, we follow the same protocol as in the LLM experiments.

**Data Collection.** Training data were collected by varying object placement over a $10 \times 10$ grid of initial positions: `obj-init-x` $\in [-0.35 - 0.12]$, `obj-init-y` $\in [-0.02, 0.42]$. For evaluation, we sampled 100 random object locations uniformly in this area.

**Supervised Fine-Tuning (SFT).** For each grid point, we collected 10 successful trajectories using the RT-1 (Brohan et al., 2022) model and filtered for successful trajectories. We trained models with batch sizes $\{16, 32, 64\}$ and learning rates $\{1\times10^{-6}, 3\times10^{-6}, 5\times10^{-6}, 7\times10^{-6}, 9\times10^{-6}, 1\times10^{-5}, 3\times 10^{-5}\}$. Other hyperparameters were: AdamW optimizer, 1 training epoch, max gradient norm of 1, weight decay of 0, warmup of 10 steps, constant-with-warmup scheduler, and `bfloat16` precision.

**Reinforcement Learning (RL).** For RL, we trained using REINFORCE with an reward normalization baseline, without explicit KL regularization. At each iteration, 5 trajectories were collected per grid point. Rewards were binary success indicators of task completion. RL training used the same training config as SFT.

## C.3 MNIST EXPERIMENTS

All MNIST experiments were conducted using a 3-layer MLP with input dimension 785, hidden layers of sizes 512 and 256, and output dimension 10. The input consisted of a flattened $28 \times 28$ image concatenated with a binary indicator: $+1$ for ParityMNIST and $-1$ for FashionMNIST.

**Pretraining.** We pretrained the network jointly on ParityMNIST and FashionMNIST using small subsets of the original datasets (500 images from each). For ParityMNIST, the label was chosen uniformly at random among all digit labels with the correct parity.

**Fine-tuning methods.** In our experiments, we evaluated five fine-tuning strategies:

1. **GRPO**.
2. **GRPO + KL regularization** with coefficient $0.1$.
3. **SFT 1**: all even digits mapped to label 0, all odd digits to label 1.
4. **SFT 2**: even digits randomly mapped to $\{0, 4\}$, odd digits to $\{1, 5\}$.
5. **SFT with oracle distribution**: annotations drawn from the minimum-KL distribution consistent with task correctness.

**Oracle distribution.** Motivated by the KL–forgetting connection, we define the oracle distribution as the one that achieves perfect task accuracy while remaining closest (in KL divergence) to the pretraining distribution $\pi_0$. Concretely, for an input image $x$ we compute $\pi_0(\cdot|x) \in \mathbb{R}^{10}$ and the binary indicator vector $R \in \{0, 1\}^{10}$ encoding which labels are correct given the digit's parity. The oracle distribution $q^*$ is the solution to:

$$q^* = \arg\min_q D_{\mathrm{KL}}(\pi_0 \| q) \quad \text{s.t.} \quad q^\top R = 1.$$

Since KL is convex and the constraint is linear, we can calculate a closed-form solution for every image. We then sample from $q^*$ to produce SFT annotations. Note that $q^*$ is the rejection sampling distribution as aforementioned in Appendix B.

**Hyperparameter sweep.** For each method we trained models across a sweep of 15 learning rates logarithmically spaced between $3e - 6$ and $1e - 3$, using either a constant-with-warmup or cosine-with-warmup scheduler, and training for 1 or 2 epochs. Including mid-training checkpoints, this produced approximately 500 runs per method.

## C.4 CENTERED KERNEL ALIGNMENT

**Centered Kernel Alignment (CKA) (Kornblith et al., 2019)** Given representations $X, Y \in \mathbb{R}^{n \times d}$, define kernels $K = XX^\top$, $L = YY^\top$. Let $H = I - \frac{1}{n}\mathbf{1}\mathbf{1}^\top$ be the centering matrix. The centered kernels are

$$\bar{K} = HKH, \quad \bar{L} = HLH.$$

CKA is then computed as

$$\mathrm{CKA}(K, L) \;=\; \frac{\langle \bar{K}, \bar{L} \rangle_F}{\|\bar{K}\|_F \, \|\bar{L}\|_F},$$

where $\langle A, B \rangle_F = \mathrm{tr}(A^\top B)$.

**CKA with $k$-NN Alignment (CKNNA) (Huh et al., 2024)**  Let $\alpha(i, j) \in \{0, 1\}$ indicate whether $i, j$ are mutual $k$-nearest neighbors in both $X$ and $Y$. Define the masked inner product

$$\langle A, B \rangle_\alpha = \sum_{i=1}^{n} \sum_{j=1}^{n} \alpha(i, j) \, A_{ij} B_{ij}.$$

CKNNA is then given by

$$\mathrm{CKNNA}(K, L) \;=\; \frac{\langle \bar{K}, \bar{L} \rangle_\alpha}{\sqrt{\langle \bar{K}, \bar{K} \rangle_\alpha \, \langle \bar{L}, \bar{L} \rangle_\alpha}}.$$

When $\alpha(i, j) = 1$ for all $i \neq j$, CKNNA reduces to standard CKA.

## D  ADDITIONAL RESULTS

### D.1  REPRESENTATION PRESERVATION

While benchmark accuracy provides an external measure of forgetting, it may conflate genuine loss of capability with superficial effects such as formatting mismatch between tasks. To assess whether fine-tuning alters the model more fundamentally, we analyzed changes to the model's representations.

**Experimental Setup.**  To study how representations change between models, we compare their embeddings on a shared dataset. Following prior work, we compare the relative geometry of the embeddings—that is, how different inputs relate to each other. This geometry can be summarized by a kernel (similarity) matrix, which encodes pairwise relationships among input embeddings. Centered Kernel Alignment (CKA) (Kornblith et al., 2019) is a standard measure for comparing such kernels, providing a way to quantify representational similarity between models.

For this analysis, we constructed kernels from random Wikipedia paragraphs, ensuring that the probe data are unrelated to the fine-tuning tasks. We then compared the kernels of the base model and its fine-tuned variants using CKNNA (Huh et al., 2024), a local-neighborhood variant of CKA (see Appendix C.4 for details). Comparisons were made between SFT and RL models that achieved similar final accuracy on the new task, isolating representational differences due to training method rather than task performance.

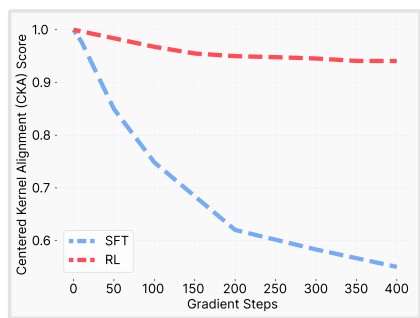

**Results.**  Figure D.1 shows that RL-trained models retain high representational similarity (CKNNA=0.94) to the base model, with CKNNA scores remaining close to one even after fine-tuning on the new task. In contrast, SFT-trained models exhibit substantial representational drift (CKNNA=0.56). These results indicate that RL fine-tuning integrates new abilities while leaving the overall representation space largely intact, whereas SFT alters the geometry more extensively. Together with the benchmark results, this suggests that RL is able to integrate new abilities without disturbing the underlying rep-

Figure 8: **CKA similarity to the base model during training.** Although SFT and RL achieve comparable task performance, SFT models diverge substantially in their representations, whereas RL models remain more closely aligned with the base model.

resentational structure, while SFT incurs representational shifts that manifest as catastrophic forgetting.

## D.2 Scaling and Forgetting

Prior work has suggested that catastrophic forgetting diminishes as model size increases (Ramasesh et al., 2021; Luo et al., 2023; Cossu et al., 2024). To evaluate this claim in our setting, we repeated the SFT experiments from Section 3 using Qwen 2.5 models with 3B, 7B, and 14B parameters on the Science Q&A task.

The results, shown in Figure 9, demonstrate that although larger models start with better general capabilities, the trade-off between new-task performance and prior-task retention remains unchanged: across all model sizes, SFT improves new-task accuracy at the expense of forgetting. In particular, to reach high accuracy on the Science Q&A task, substantial degradation occurs in performance on prior benchmarks regardless of model scale.

## D.3 Optimization Dynamics

To examine the link between parameter updates and forgetting, we analyzed the optimization trajectory at the level of individual training steps. For each update, we computed two quantities:

Figure 9: Pareto frontiers for SFT on Qwen 2.5 Instruct models of size 3B, 7B, and 14B on the Science Q&A task. All sizes exhibit the same fundamental trade-off—gains on the new task require forgetting prior capabilities.

1. **Forgetting direction.** Using the FashionMNIST evaluation set, we calculated the gradient of the loss with respect to model parameters. We then measured the cosine similarity between this gradient and the actual parameter update from the training step. A positive cosine indicates that the update increases FashionM-NIST loss (catastrophic forgetting), while a negative cosine indicates an update that reduces it.

2. **KL shift.** We measured the change in KL divergence between the model's output distributions on the Pari-tyMNIST test set before and after the update.

Plotting per-step KL change against the cosine similarity (Figure 11) revealed a strong correlation: steps producing larger KL shifts tended to align more with the forgetting gradient. This analysis demonstrates that at the level of optimization dynamics, catastrophic forgetting is driven by updates that induce larger distributional shifts on the new task.

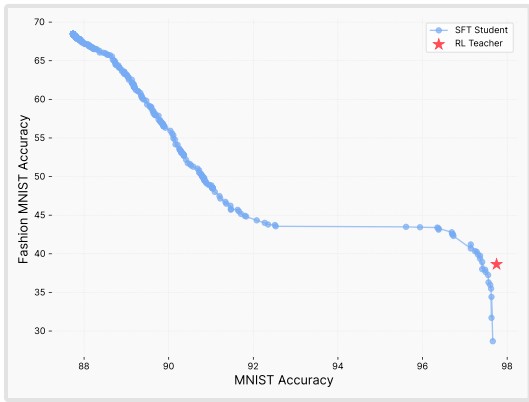

Figure 10: **SFT distillation from an RL teacher.** Accuracy trade-off between the new task (MNIST) and the prior task (FashionMNIST). Sweeping student hyperparameters shows that SFT can match the teacher within noise on both tasks. This suggests that what matters is not the optimization path, but the distribution of the final model.

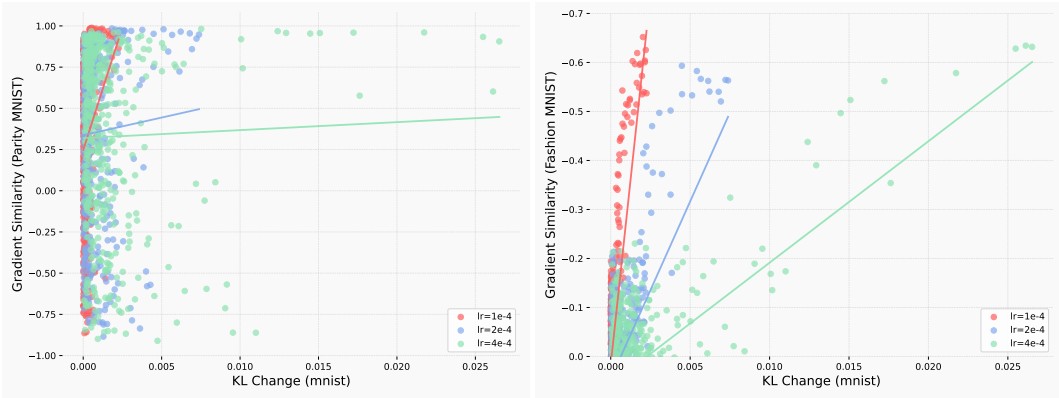

Figure 11: **Gradient similarity versus KL change.** (Left) On the new training task (ParityMNIST), gradient cosine similarity and KL change per step remain uncorrelated. (Right) On the prior task (FashionMNIST), the gradient similarity is more correlated with the KL change per step on the training task (ParityMNIST). Together, these plots show that taking a larger step on the current task induces gradients that are more similar in direction that forgets the most.

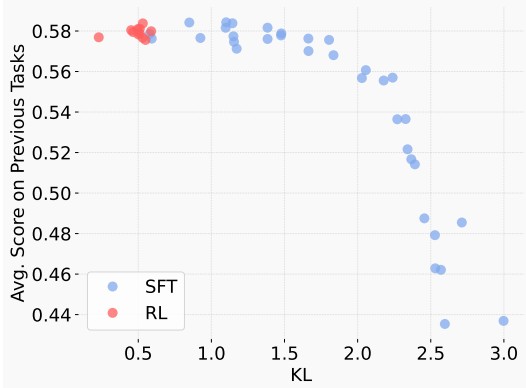

Figure 12: We plot the KL divergence between the base and fine-tuned model on the new task, alongside the corresponding forgetting performance across methods.

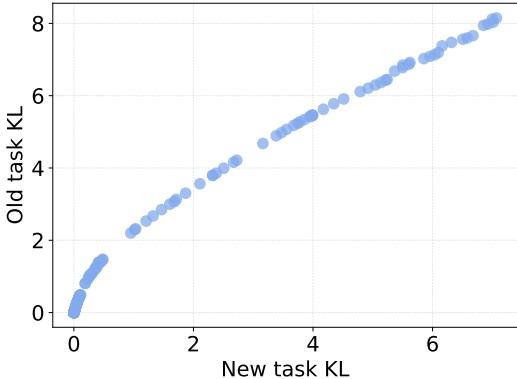

Figure 13: KL between the base and trained policy on new task is highly correlated to the KL on old task.

