# OpenReview forum: "RL's Razor: Why Online Reinforcement Learning Forgets Less"
_ICLR.cc/2026/Conference — ICLR 2026 Poster_

### Official Review · Reviewer_xH9f · 2025-10-26

**Soundness:** 2
**Presentation:** 3
**Contribution:** 2
**Rating:** 6
**Confidence:** 4

**Summary:**

This paper finds that fine-tuning base model by RL on new tasks can better resist catastrophic forgetting compared to fine-tuning by SFT. The authors find that the KL divergence between fine-tuned model and base model can be a strong predictor of catastrophic forgetting. RL forgets less because on-policy sampling inherently biases the model toward KL-minimal solutionns, while SFT may drift arbitrarily far to adapt to new tasks. Their empirical evidence and theoretical justifications support the claims.

**Strengths:**

1. The cor insight is simple but efficient: the extent of catastrophic forgetting is determined by KL divergence. That RL does better than SFT is because the on-policy nature which minimizes the KL divergence. This insight can bring more insights of RLHF and continual post-training research
1. The takeaway messages are consice and clear, directly conveying the most important conclusions.
1. The experiments including LLM and robotics control are comprehensive to illustrate the effectiveness and generality of the KL-based explanation, demonstrating consistent forgetting behavior and RL's advantage.

**Weaknesses:**

1. The analysis of KL divergence in catastrophic forgetting is still high level:
    1. The sampling distribution and KL divergence is a correlation analysis but not a causality explanation. A mathematics proof should be more helpful to support this finding
    1. The paper does not provide a deeper examination of how KL drift translates into the loss on prior tasks. For instance, it is unclear whether the degradation arises from representational collapse on past tasks, or from changes in critical neurons that significantly affect previously acquired skills. A more fine-grained analysis would greatly strengthen the causal argument behind the proposed explanation.
1. As KL divergence is an indicator to forgetting, it would be meaningful to add KL-regularized online and offline methods as baselines, e.g.,[1,2,3], which also extends the scope if minimizing KL divergence directly can mitigate forgetting.
1. The experiments in the paper indicate that the forgetting is substantial in toy experiments, but noticeably milder in LLM-based tasks. However,Fig. 2 and Fig. 4 exaggerate the performance loss by compressing the y-axis. This plotting is misleading and give readers an inflated perception of catastrophic forgetting, overstating the empirical impact of the proposed phenomenon.

[1] Azar, Mohammad Gheshlaghi, et al. "A general theoretical paradigm to understand learning from human preferences." International Conference on Artificial Intelligence and Statistics. PMLR, 2024.

[2] Ethayarajh, Kawin, et al. "Kto: Model alignment as prospect theoretic optimization, 2024." URL https://arxiv. org/abs/2402.01306.

[3] Wu, Yifan, George Tucker, and Ofir Nachum. "Behavior regularized offline reinforcement learning." arXiv preprint arXiv:1911.11361 (2019).

**Questions:**

1. In the toy experiment, forgetting is significant, but in LLM tasks, the absolute value of forgetting is small. Why is there such a big gap between the two? Is it due to differences in model size, optimization robustness, task overlap, or KL sensitivity across models of different sizes?
1. The compression in Fig. 2 and Fig. 4 visually makes forgetting look more severe than the actual numerical value.  Would the author be willing to provide an absolute difference or a version with a uniform scale?
1. The authors test alternative indicators in distribution change, weight change, and activation change in Sec. 6. Have the authors tested other distribution-level indicators, like JS divergence and Wasserstein distnace? Will they perform better?
1. Is this finding only suitable for on-policy RL, what about off-policy RL?

---

> ### Author Response · Authors · 2025-11-24
> **Response to Reviewer xH9f (Part 1)**
>
> We thank the reviewer for their constructive feedback and for highlighting the strengths of our work, including the clarity of our core insight, the comprehensive empirical evaluation across LLMs and robotics, and the potential implications for RLHF and continual post-training research. We address all concerns and questions below.
>
> > The sampling distribution and KL divergence is a correlation analysis but not a causality explanation. A mathematics proof should be more helpful to support this finding
>
> We appreciate this point. We acknowledge that the connection between KL drift and forgetting found in this paper is a correlation, not a causal connection. Establishing a stronger causal and mechanistic connection is a difficult open research question; we explicitly state this limitation in the paper However, we hope that this work will encourage the community to look into this question in the hope it will lead to new and useful insights on continual learning.
>
> >  it is unclear whether the degradation arises from representational collapse on past tasks, or from changes in critical neurons that significantly affect previously acquired skills.
>
> We would like to direct the reviewer to Appendix D.1. There, we explicitly analyze representation drift across layers during fine-tuning. We find that SFT induces significant shifts in internal representations, whereas RL produces only minimal changes. Moreover, in the SFT setting (where the KL shift is large), the representation drift appears across all layers rather than being concentrated in a small set of “critical neurons.” This suggests that the degradation is not due to the disruption of a few key units, but rather reflects a broad reorganization of the model’s feature space.
>
> > it would be meaningful to add KL-regularized online and offline methods as baselines,
>
> We thank the reviewer for these suggestions. We do test SFT + KL regularization (Figure 6) and show that explicit KL regularization helps RL but has only a marginal effect for SFT. Based on the reviewer's suggestion, we also ran an IPO experiment on the Science Q&A task, and found that the results are consistent with our previous findings and theory - the KL term does not able to mitigate forgetting.
>
> https://anonymous.4open.science/api/repo/rl_razor_rebuttal-5972/file/ipo.pdf?v=66a54a77
>
> > The experiments in the paper indicate that the forgetting is substantial in toy experiments, but noticeably milder in LLM-based tasks. However,Fig. 2 and Fig. 4 exaggerate the performance loss by compressing the y-axis. This plotting is misleading and give readers an inflated perception of catastrophic forgetting, overstating the empirical impact of the proposed phenomenon.
>
> Thank you for this observation. We observe more forgetting in the toy experiments because we can train both SFT and RL fully to convergence. In the LLM setting, however, we cannot train to convergence due to the substantial compute required. With more compute, we expect the forgetting curves to show larger absolute drops. Even so, the percentage-point degradation from SFT remains meaningful in the current experiments, and the relative gap between SFT and RL is consistent across settings.
>
>
> > In the toy experiment, forgetting is significant, but in LLM tasks, the absolute value of forgetting is small. Why is there such a big gap between the two? Is it due to differences in model size, optimization robustness, task overlap, or KL sensitivity across models of different sizes?
>
> This is an interesting question, and it is due to the lack of training iterations we did in the LLM setting due to compute limits. When we analyze the various SFT checkpoints we got in our experiments, we do see that the more gradient steps you take, the more forgetting (and KL) you accumulate. Our datasets are just relatively small compared to the size of the models, and therefore there is only a limited amount of forgetting happened.

---

> > ### Author Response · Authors · 2025-11-24
> > **Response to Reviewer xH9f (Part 2)**
> >
> > > The compression in Fig. 2 and Fig. 4 visually makes forgetting look more severe than the actual numerical value. Would the author be willing to provide an absolute difference or a version with a uniform scale?
> >
> > We will revise the text to highlight the numerical value. We wonder if the reviewer finds this absolute difference version better: https://anonymous.4open.science/api/repo/rl_razor_rebuttal-5972/file/catastrophic_science%20(1).pdf?v=b5c527c7
> >
> > > The authors test alternative indicators in distribution change, weight change, and activation change in Sec. 6. Have the authors tested other distribution-level indicators, like JS divergence and Wasserstein distnace? Will they perform better?
> >
> > We thank the reviewer for this suggestion. As shown in Table 1, we evaluated a wide range of distribution-level metrics, including JS-related measures (via forward and reverse KL), total variation, L2 distribution change.
> >
> > > Is this finding only suitable for on-policy RL, what about off-policy RL?
> >
> > This is a great question. Currently using Q learning to train foundation models (either LLMs of VLAs) is not an established practice, and the right way to do it is stll an open research question. Therefore, we limited our study to on-policy methods.
> >
> > We hope this addresses the reviewer’s questions and concerns, and we look forward to engaging further to continue improving the work.

---

### Official Review · Reviewer_muvc · 2025-10-27

**Soundness:** 4
**Presentation:** 4
**Contribution:** 3
**Rating:** 8
**Confidence:** 4

**Summary:**

The paper investigates why reinforcement learning (RL) fine-tuning tends to forget less than supervised fine-tuning (SFT) when adapting foundation models to new tasks. Across LLM and robotics benchmarks (and a controlled ParityMNIST toy experiment), the authors show an empirical forgetting law: the degree of catastrophic forgetting is well predicted by the KL shift measured on the new-task inputs, $\mathbb{E}_{x\sim\tau}\big[ \mathrm{KL}(\pi_0\|\pi)\big]$. They argue (and empirically validate) that on-policy RL is implicitly biased toward KL-minimal solutions among all policies that solve the new task — because RL samples from and reweights the model’s own outputs — while SFT can converge to arbitrarily distant solutions given the annotator distribution.

The paper’s contributions are threefold: (1) empirical demonstration that RL fine-tuning preserves prior capabilities better than SFT at matched new-task performance, (2) the KL–forgetting law showing forward (and backward) KL on the new-task distribution predicts forgetting, and (3) theoretical and experimental evidence for “RL’s Razor” — a proof that policy-gradient on-policy methods converge to the optimal representable policy closest in KL to initialization, plus ablations (oracle SFT, KL regularization, different objectives) that validate the mechanism.

**Strengths:**

The paper presents solid evidence for a novel and significant claim that online RL finetuning preserves the base model's knowledge better than SFT finetuning. This claim has far-reaching consequences for fine-tuning pipelines in foundational models, both in LLMs and Robotics (VLAs and Behavioral Foundation Models) space. In addition to an extensive empirical evaluation of SFT and RL finetuning on Qwen 2.5 models and OpenVLA, the paper also presents the forgetting phenomenon during fine-tuning from a theoretical perspective. I find the provided toy experiment on ParityMNIST to be illustrative and instructive for gaining intuition. The soundness of the experiment section is high-level, and the presentation is clear and of high quality.

**Weaknesses:**

The paper does not present any obvious weaknesses.

**Questions:**

One clarity remark - I am not sure how dist. 1 and dist. 2 differ - it's not easy to find in the text or in the Figure caption.
Additionally, I would like to know whether the authors plan to conduct any further experiments with VLA, as the paper currently only presents a single task: Pick and Place. I'm wondering whether the trend of catastrophic forgetting with SFT is more severe across some family of manipulation tasks and maybe less severe in some others? Can authors provide any intuition about that?

---

> ### Author Response · Authors · 2025-11-24
> **Response to Reviewer muvc**
>
> We thank the reviewer for their thoughtful and positive assessment of our work. We appreciate their recognition of the novelty and significance of our central claim as well as their comments on the clarity of our presentation and the soundness of our empirical and theoretical analyses. We are especially glad that the ParityMNIST toy experiment was found to be intuitive and instructive, and that the evaluation across LLMs and VLAs was viewed as solid and high-quality. We address the reviewer’s remaining questions below.
>
> > One clarity remark - I am not sure how dist. 1 and dist. 2 differ - it's not easy to find in the text or in the Figure caption.
>
> We thank the reviewer for pointing out the lack of clarity regarding Dist. 1 and Dist. 2. These refer to the two annotator distributions used for SFT in the ParityMNIST experiment (Appendix C.3). We will update both the main text and the figure captions to make this explicit. For clarity, the two distributions are:
>
> Dist. 1: all even digits are mapped to label 0, all odd digits to label 1.
>
> Dist. 2: even digits are randomly mapped to {0, 4}, and odd digits to {1, 5}.
>
> >  Additionally, I would like to know whether the authors plan to conduct any further experiments with VLA, as the paper currently only presents a single task: Pick and Place.
>
> We are unable to include further robotics experiments during the rebuttal stage. However, we believe the Pick-and-Place results are informative precisely because OpenVLA-7B has been extensively pre-trained on many variations of pick-and-place trajectories. Since catastrophic forgetting appears even on this well-represented manipulation family, we expect it to be at least as severe, if not more so, on manipulation tasks that are less common in the training distribution. This is because learning them through SFT will result in even bigger KL deviation, and thus more forgetting.
>
> We hope this addresses the reviewer’s questions and concerns, and we look forward to engaging further to continue improving the work.

---

### Official Review · Reviewer_8HAS · 2025-11-01

**Soundness:** 2
**Presentation:** 2
**Contribution:** 2
**Rating:** 2
**Confidence:** 4

**Summary:**

The paper proposes the hypothesis that RL finetuning, compared to supervised finetuning leads to a minimal change (in the KL sense) in the model distribution, therefore minimizes forgetting. They partially attribute this behavior to the success of RLHF for large LLM finetuning.
The authors provide both empirical evidence on LLMs (Math reasoning, Science Q&A, Tool use) and a robotic task (Pick and Place) as well as on a toy task build out of MNIST and FashionMNIST that they call parityMNIST. Finally they formalize and provide some theoretical justification of their hypothesis.

Overall I think the hypothesis is interesting. On the other hand I feel the paper is quite liberal in the claims that it makes and potentially (and most likely not intentionally) misinterprets some of the previous work.

**Strengths:**

To me the main strength of the work is the hypothesis proposed. I think it could be the basis of a potentially a novel perspective, and provides an angle on the discussion between RLFT and SFT that I have not seen yet. I'm not aware of work looking at whether on-policyness affects catastrophic forgetting.  The paper provides some decent arguments in favor of their perspective, both empirically and theoretically, though I'm not sure if these arguments are sufficient.

**Weaknesses:**

My main concern is probably with the generality of the claims (and potentially the lack of ablation to be sufficiently convincing). I would have preferred everything to be a bit more contextualised. Maybe the RL Razor hypothesis (not sure if it is the best name for the hypothesis) is fine -- which I understand it as the source of less forgetting being on-policy behaviour of RL compared to SFT.

But the empirical results are considerably more specific (e.g. it is about comparing GRPO in the RL context with AdamW for the SL context). Generally I think the main body of the paper is light on technical details (pushing many of these to the appendix), to a point that is harming the scholarly understanding of the work, brushing away some of the nuance of the results. I also think that the paper has not done sufficient work to isolate or ablate in order to show that it is the on-policyness that leads to this behaviour rather than other aspects (e.g. the regularization that is part of the GRPO objective which are also typical for certain optimization algorithms in SL).

In terms of generality of the claims:  the choice of optimizer will greatly affect how far away you travel from the starting point. Is not clear to me that a different choice (e.g. natural gradient that includes a KL constraint not that different from GRPO/PPO but enforced more properly) would not lead to a similar minimal change in policy. Of course the approximation made to make the algorithm run efficiently can drastically impact things, and I can not tell if it would be equivalent or not. But that is the problem. The paper makes a broad argument about supervised learning, implying that this will be true for any choice of optimizer.  The same on the RL side, change in objective, change on the optimizer used in the RL setting will of course drastically change how much the model will traverse in functional space making it impossible to predict if the two settings will rank the way the authors are saying. So please be specific in claims.

In terms of ablations: GRPO contains a regularization as well as a clipping component (beside maximizing the likelihood of the action that gives bigger reward). Is not clear to me to what extent these components are the one driving the observed behaviour, compared to the argument made by the authors that it is because things are on policy. Specifically we know that PPO for example can diverge in RL settings, which to me is the opposite of doing minimal change in the policy in the KL sense. Actually a lot of RL algorithms suffer from such instabilities. How are these behaviours integrated in this hypothesis?

Theory: While I understand the argument made for the RL algorithm, I'm not sure enough work has been put in the counterpart argument, that SFT does not reach a minimal change in the KL sense. Am I misunderstanding this? There does not seem to be any Lemma in the main paper saying that SFT does not do the same as RL, and is not clear to me that bar the optimizer used, SFT would not do the same as RLFT.

I have two main issues with the theory:
 (1) It does not take into account the function approximator (i.e. the neural network) that I think can easily invalidate the argument. I understand that the neural network makes everything non-linear and hence any explicit/concrete mathematical argument is hard to make. But at least acknowledge this. Obviously due to shared parametrization, when the loss is minimized on a particular set of observation, this can change in arbitrary ways the behaviour of the model on not observed data (which is the main point of catastrophic forgetting). So while I appreciate the attempt of adding some theoretical results, I'm not sure to what extent they speak in favor of the hypothesis.

As a said note, a lot of continual learning work argues that is specifically the neural network (and its non-linear nature) that makes catastrophic forgetting interesting.

(2) Maybe related to the above point. If you consider the tabular case. I do not see who being on policy vs not (in the case where the optimal policy is achievable) leads to different solutions. Maybe there is something to the argument that I'm missing.

**Questions:**

1. Can you justify why not the KL constraint embedding in RL objective to stay close to the previous policy is not the main reason why these methods converge to a solution that has a minimal KL change? E.g. jumping to the appendix (lines 805-809) you say that for SFT equals the minimum-KL optimum policy. Why is that the case? What is the technical argument? In my view it should lead to a minimum -KL policy if that is representable by pi for any beta, no? Actually where is the proof of theorem A.1?

2. There is no justification of why the KL computed on the new task data is sufficient to measure forgetness. Can you say more about this (particularly as I feel that implicitly in the later part of the paper you are implicitly making the claim that the fact that that CL algorithms look at the KL on the old data is somewhat problematic -- if that is not the intent, could you have a discussion of this in the paper and make your view explicit). I think it is easy to provide an argument of why KL on new data does not have to correlate with forgetness. E.g. if the old task and the new task have different supports, this metric will say very little about how much you forget (at least under a standard definition of forgetting). So it seems is more of an instance that the old task and new task are sufficiently similar (for an unknown definition of similar). Please also see the connection of this choice and algorithms like Learning without forgetting: https://arxiv.org/abs/1606.09282 which is not cited in current work.  In LwF paper the choice of computing on new data is justified from being cheaper (requiring not to store data) but no claim is being made that is more justified. But I think the predominant view is that the KL computed in the old data tracks the quantity we care, so if the paper tries to argue against this, it should be made more explicit and there should be more evidence for the it.

3. I would rephrase a bit section 6. Actually similar claims are made in the intro. E.g. Kirkpatrick et al do not aim at minimizing change in parameter space. The objective of EWC is explicitly in functional space and talks about the KL term similar to the one you mention (except computed on the old data, so if the claim is that the particular choice of data is wrong please clarify and explain why). Most of CL works do this.  The approximation used to instantiate the algorithm leads to a parameter level regularization due to taking a second order taylor expansion of the KL term (and then approximating the Fisher by a diagonal). But in spirit most of CL algorithms explicitly target minimizing the change in functional space. It is just that minimizing this quantity it needs to be approximated by something cheaper. The framing currently sounds as if these works are minimizing the change in parameters hoping that will lead to less forgetting without connecting to the KL term or acknowledging it.  I would also argue that the claim that these metrics correlate weakly with the KL change is a bit misleading. They are direct approximations of the KL, and for sufficiently large models (where second order Taylor holds), they correlate quite well. Please provide evidence otherwise (and extend the technical description of what exactly is being run for e.g. for table 1).

4. Can you actually have comparisons with at least some basic CL methods? At least replay, which is sometimes used with SFT. Or with SFT + KL regularization?

5. Can you ablate further the RL algorithm used (e.g. PPO, reinforce) or rather remove terms of the GRPO objective to ablate what part of the algorithm empirically leads to small KL term (is the on-policyness or sampling from the model distribution, is it the clipping or KL regularization).

6.  Can you provide some comments regarding the theory part (outlined above)? I.e. discussing the tabular case, or at least discussing the role of the function approximator in this.

**Details Of Ethics Concerns:**

I do not see the need of an ethic review for this work.

---

> ### Author Response · Authors · 2025-11-24
> **Response to Reviewer 8HAS (Part 1)**
>
> We appreciate the reviewer’s constructive feedback and are glad they found the hypothesis interesting. We addressed their concerns below.
>
>
> > I also think that the paper has not done sufficient work to isolate or ablate in order to show that it is the on-policyness that leads to this behaviour rather than other aspects… GRPO contains a regularization as well as a clipping component (beside maximizing the likelihood of the action that gives bigger reward). Is not clear to me to what extent these components are the one driving the observed behaviour
>
> The reviewer raises an important point and contemplates whether clipping and KL regularization in GRPO may explain the phenomenon observed in the paper. However, our implementation of GRPO did not use these mechanisms. In line 185 of our paper, we explicitly mention that no KL regularization was used. Regarding clipping, we use GRPO with only a single gradient step per data generation step, $\pi_{\theta_{old}}$ is identical to $\pi_{\theta}$, reducing the update to standard REINFORCE with normalized rewards. The 1-0 REINFORCE experiment (Figure 4) removes even the reward normalization, thus demonstrating that it is only the on-policy nature of RL that prevents it from forgetting.
>
> We acknowledge that this was not made sufficiently clear in the main text, and many details were only in the appendix. We will revise the manuscript to highlight this explicitly.
>
> > the choice of optimizer will greatly affect how far away you travel from the starting point. Is not clear to me that a different choice would not lead to a similar minimal change in policy.
>
> The reviewer also notes that our results can be explained by the use of AdamW optimizer. To rule out dependence on particular choice of optimizer, we conducted additional ablations using RMSProp and AdaGrad, and found that our conclusions remain unchanged across optimizers. Although some optimizers are able to push the model more or less than others, the learning-forgetting trade-off remains the same (see https://anonymous.4open.science/api/repo/rl_razor_rebuttal-5972/file/Optimizers.pdf?v=253f8e54
> ). We will update the main text to reflect this optimizer-agnostic behavior. We thank the reviewer for pointing this out, as it makes our empirical results even stronger.
>
>
> > While I understand the argument made for the RL algorithm, I'm not sure enough work has been put in the counterpart argument, that SFT does not reach a minimal change in the KL sense.
>
> The reviewer is correct that our theoretical analysis focuses on the RL side, and we appreciate the chance to clarify why an analogous treatment for SFT is not presented. Our core argument is the following: in on-policy RL, the training distribution is self-generated by the current policy. This allows us to characterize RL’s implicit bias toward KL-minimal solutions in a clean and general way.
> In contrast, SFT learns from an externally supplied dataset whose distribution can be arbitrarily far from the base model. Because the KL divergence depends directly on the training distribution, there is no universal guarantee that SFT will converge to the minimum-KL optimal solution. Some SFT datasets may by coincidence lead to the minimal-KL solution, but many will not and there is no distribution-agnostic theoretical criterion that ensures SFT will do so.
> Empirically, we find exactly this behavior. Typical SFT datasets (as the ones in our LLM and Robotics experiments) do not induce minimum-KL solutions. However, in toy problems (such as MNIST) we were able to synthetically construct a KL-minimal SFT dataset (Figure 3) that achieves an almost ideal learning–forgetting tradeoff, even surpassing RL.

---

> > ### Author Response · Authors · 2025-11-24
> > **Response to Reviewer 8HAS (Part 2)**
> >
> > > I have two main issues with the theory: (1) It does not take into account the function approximator (i.e. the neural network) that I think can easily invalidate the argument.
> >
> > We appreciate the reviewer’s point regarding the role of the function approximator. Our theoretical analysis intentionally uses a simplified convex setting to isolate the effect of on-policy sampling, rather than attempting to fully characterize the optimization dynamics of deep neural networks. While the proof is restricted to convex model classes, the underlying intuition does not rely on convexity: the key requirement is that the model class is realizable and that optimization converges to a global optimum. Recent overparameterization results suggest that large neural networks often satisfy these assumptions [1,2]. However, extending our analysis to fully nonlinear networks would require substantial additional machinery and is outside the scope of this work.
> >
> > Our goal is not to claim a complete theory of catastrophic forgetting in deep networks, but to highlight a general principle - on-policy RL with binary rewards induces a type of iterative optimization, where the policy repeatedly trains on its own sampled distribution. This self-generated data distribution makes the update behave as an implicit projection toward the minimum-KL optimal solution. Understanding the precise convergence conditions for modern neural architectures is an important direction for future work; however, we do not expect such an extension to alter the paper's main conceptual takeaway.
> >
> > [1] Du, Simon S., et al. "Gradient Descent Provably Optimizes Over-parameterized Neural Networks." International Conference on Learning Representations 2019.
> >
> > [2] Du, Simon, et al. "Gradient descent finds global minima of deep neural networks." International conference on machine learning 2019.
> >
> > > (2) Maybe related to the above point. If you consider the tabular case. I do not see who being on policy vs not (in the case where the optimal policy is achievable) leads to different solutions. Maybe there is something to the argument that I'm missing.
> >
> > In the tabular setting, suppose we start from a base policy and define the new task as a set of state–action pairs with a reward of 1. In this case, RL is guaranteed to reach min KL solution, while SFT does not:
> >
> > With on-policy RL, updates are taken only on actions that already have support under the base policy. As a result, the policy improves by reweighting the base policy’s own action distribution, which naturally keeps it close to the original policy in KL. In other words, among all optimal solutions, on-policy RL tends to converge to the one closest to the base policy (as illustrated in Fig. 5).
> >
> > SFT has no such constraint. Its behavior is entirely determined by the data distribution of the SFT dataset, which may correspond to any optimal solution—including one arbitrarily far in KL from the base policy. Nothing in the SFT objective prefers the KL-minimal solution; it simply imitates whatever labels it is given.
> >
> > Thus, even in the tabular case, the two training procedures generally converge to different optimal policies: RL to the KL-minimal one, SFT to whichever one is encoded by the demonstrations. This difference is exactly what explains the difference in forgetting behavior.
> >
> > > Can you justify why not the KL constraint embedding in RL objective to stay close to the previous policy is not the main reason why these methods converge to a solution that has a minimal KL change?
> >
> > As mentioned earlier, we do not employ any explicit KL regularization in our RL experiments, so the minimal-KL behavior we observe does not stem from a KL penalty; we will revise the main text to clarify this point.
> >
> > >  jumping to the appendix (lines 805-809) you say that for SFT equals the minimum-KL optimum policy. Why is that the case?
> >
> > Regarding the reviewer’s question about SFT and Theorem A.1: in Appendix (lines 805–809), we construct a special oracle annotator distribution that is both reward-optimal and the minimum-KL solution, and in this specific case, standard cross-entropy minimization ensures that SFT recovers that minimum-KL policy. In contrast, Theorem B.8 shows that for a general annotator q, SFT with KL regularization does not recover the minimum-KL optimal policy for any β > 0. The full proofs are presented in Appendix B, and we will clarify this distinction further in the revision.

---

> > > ### Author Response · Authors · 2025-11-24
> > > **Response to Reviewer 8HAS (Part 3)**
> > >
> > > > There is no justification of why the KL computed on the new task data is sufficient to measure forgetness.
> > >
> > > Measuring forgetting using the KL divergence on prior-task data is indeed the most direct and accurate predictor of forgetting, since it evaluates distributional drift exactly where forgetting occurs. However, for modern foundation models that undergo many rounds of post-training (each using different datasets, objectives, etc.), it is typically infeasible to reconstruct or access the full distribution of prior tasks needed to compute this KL. This practical limitation motivated our focus on KL measured on the new task, and our finding that it provides a robust and reliable proxy for forgetting. We agree that this motivation was not made sufficiently explicit in the paper, and we will revise the paper to clarify this point.
> > >
> > >
> > > >  I think it is easy to provide an argument of why KL on new data does not have to correlate with forgetness. E.g. if the old task and the new task have different supports
> > >
> > > Regarding the concern about disjoint supports, we empirically find that even when the old and new tasks are constructed to have completely non-overlapping input distributions, the KL divergence measured on the model outputs given new task inputs still tracks the KL given the prior task inputs well. This is demonstrated in our Parity-MNIST experiment (plot: https://anonymous.4open.science/api/repo/rl_razor_rebuttal-5972/file/old_new_kl.pdf?v=844de1fd), where the two tasks share no input support by design. The underlying reason is that in neural networks, parameter updates induced by training on one task propagate through shared layers and therefore influence predictions on all inputs, not just those from the new task. As a result, KL drift measured on the new task captures the magnitude of global parameter change, which in turn affects performance on the old task. Thus, even with disjoint supports, the KL on new-task data can still serve as a reliable proxy for forgetting in practice.
> > >
> > > > Please also see the connection of this choice and algorithms like Learning without forgetting: https://arxiv.org/abs/1606.09282 which is not cited in current work.
> > >
> > > Thank you for pointing us to Learning without Forgetting. Similar to EWC, this method can be understood as approximating a KL penalty on the new-task data to mitigate forgetting. We will add it to the related-work section. Our work differs from these approaches in that we show how on-policy RL implicitly performs KL regularization, without requiring an explicit approximation or additional loss term.
> > >
> > > > I would rephrase a bit section 6. Actually similar claims are made in the intro. E.g. Kirkpatrick et al do not aim at minimizing change in parameter space.
> > >
> > > We thank the reviewer for this clarification and agree that our current phrasing in Section 6 (and parts of the introduction) may be misleading. You are correct that EWC derives its objective as an approximation to the KL between the old and new policies, as we discussed in lines 141–149. However, there are other works that operate purely in parameter space without any connection to KL approximations. To avoid conflating these approaches, we will revise Section 6 to remove the reference to EWC in this context and focus only on methods that do not explicitly relate their parameter-space penalties to KL-based reasoning.
> > >
> > > > Can you actually have comparisons with at least some basic CL methods? At least replay, which is sometimes used with SFT. Or with SFT + KL regularization?
> > >
> > > We did run experiments for SFT + KL regularization, as shown in Figure 6. While adding a KL penalty meaningfully improves retention for RL, it has only a marginal effect on SFT—the SFT + KL curve closely tracks the standard SFT curve. Thus, KL regularization does little to improve the SFT learning–forgetting tradeoff. We also provide a theoretical explanation for that in Appendix A.
> > >
> > > We hope this addresses the reviewer’s questions and concerns, and we look forward to engaging further to continue improving the work.

---

### Official Review · Reviewer_Qf7W · 2025-11-03

**Soundness:** 3
**Presentation:** 3
**Contribution:** 3
**Rating:** 8
**Confidence:** 4

**Summary:**

This paper shows the effectiveness of using RL in post training in terms of the catastrophic forgetting. Especially, compared to using RL, post-training the model using SFT suffers from forgetting when the model tries to adapt to new tasks. Through extensive experiments, the authors show that using on-policy RL makes the model converge to much closer point from the base model, and using SFT can converge to the points far from the base model. In the experiments on analyzing the contributions of sampling distribution and negative examples to forgetting, the interesting point is that on-policy sampling mainly contributes to preventing the model from being apart from the base model.

**Strengths:**

1. The main contribution on analyzing the RL in terms of the catastrophic forgetting is quite novel. Most of the experiment results are well-aligned to support the claim on the effectiveness of RL, and proposing the quantitative metric on measuring the forgetting has significant contribution.

**Weaknesses:**

I think this paper is well-written, and cannot find the significant weaknesses.

1. It would be better to specify that all of the RL approaches (e.g. RLVR or RLHF) are robust to the forgetting, and such robustness is still occurs when the model hacks the reward during RL training.

2. It would be better to show the results in terms of the model sizes. I know training much larger LLMs takes enormous resources even though using LoRA tuning, but I still wonder the larger model is also robust to the forgetting.

**Questions:**

1. Does this phenomenon also occurs in multi-trun RL? (e.g interaction with environments in agentic RL)

2. Does the robustness come from the on-policy sampling or clipping in PPO-like on-policy methods?

---

> ### Author Response · Authors · 2025-11-24
> **Response to Reviewer Qf7W**
>
> We appreciate the reviewer’s constructive comments and their recognition of our novel contribution to understanding catastrophic forgetting.
>
> > It would be better to specify that all of the RL approaches (e.g. RLVR or RLHF) are robust to the forgetting, and such robustness is still occurs when the model hacks the reward during RL training.
>
> This is an important point, and we will revise the manuscript to more clearly communicate that online RL is a broad umbrella encompassing a range of popular methods, including RLVR and RLHF.
>
> > It would be better to show the results in terms of the model sizes.
>
> We fine-tuned 3B, 7B, and 14B models using SFT (Figure 9) and observed that larger models follow a similar Pareto-frontier forgetting curve, but with the curve shifted upward. All the curves exhibit the same relative drop in the prior tasks, indicating that even as models become larger, they remain vulnerable to forgetting.
> We were unable to replicate this experiment with RL due to the substantially higher compute requirements. However, even at the 3B scale tested, RL already exhibited less forgetting than the 14B SFT model.
>
> > Does this phenomenon also occurs in multi-trun RL?
>
> This is an interesting question. While we do not currently have multi-turn RL experiments, we see no reason why the phenomenon would not extend to the multi-turn setting, provided the RL method is on-policy.
>
> > Does the robustness come from the on-policy sampling or clipping in PPO-like on-policy methods?
>
> We thank the reviewer for this important question. In fact, we do not use clipping in our experiments. The robustness we observe arises from online on-policy sampling: because we use GRPO with only a single gradient step per data generation step, $\pi_{\theta_{old}}$ is effectively identical to $\pi_{\theta}$, reducing the update to standard REINFORCE with normalized rewards. We acknowledge that our writing did not make this clear, and we will revise the manuscript accordingly.
>
> We hope this addresses the reviewer’s questions and concerns, and we look forward to engaging further to continue improving the work.

---

### Meta-Review · Area_Chair_B4xo · 2026-01-07

**Summary:**

## Overview

This paper investigates catastrophic forgetting in foundation model fine-tuning, demonstrating that reinforcement learning preserves prior knowledge significantly better than supervised fine-tuning even when achieving comparable new-task performance. The authors propose an empirical forgetting law: the degree of forgetting is predicted by KL-divergence between the fine-tuned and base policy measured on the new task. They introduce ``RL's Razor''---the principle that on-policy RL implicitly converges to KL-minimal solutions among policies that solve the new task. The work includes experiments on LLMs (Qwen 2.5) and robotic models (OpenVLA), a controlled ParityMNIST toy problem, and theoretical analysis in a simplified convex setting.

The submission received ratings of 8, 8, 6, and 2, reflecting substantial disagreement among reviewers. After carefully reviewing the paper, all reviewer comments, and the authors' responses, I believe this work makes a valuable contribution that merits acceptance, though several presentation issues must be addressed.

## Assessment

### Strengths

This paper makes a genuine and timely contribution to understanding catastrophic forgetting in foundation models. I find several aspects particularly strong:

**Novel perspective with practical impact.** The core insight---that on-policy RL's self-generated training distribution naturally biases it toward KL-minimal solutions---is both novel and practically important for foundation model deployment. The empirical forgetting law (KL-divergence as predictor) is actionable and gives practitioners something concrete to monitor during training.

**Elegant experimental design.** The ParityMNIST experiment is genuinely clever. By constructing an oracle SFT distribution that provably minimizes KL, the authors demonstrate that RL's advantage comes from implicit KL minimization rather than something inherent to the RL objective itself. When SFT uses this oracle distribution, it matches or exceeds RL's performance. This is convincing evidence for the proposed mechanism.

**Comprehensive empirical validation.** Experiments span multiple domains (LLM math reasoning, science Q\&A, tool use, and robotic pick-and-place) with consistent findings. The robustness across settings strengthens the claims considerably. The systematic ablations in Section 6 rule out alternative explanations (weight-level changes, representation drift, sparsity, rank).

**Theoretical grounding.** While restricted to simplified settings, Theorem 5.2 provides formal justification for why policy gradient converges to KL-minimal solutions in the binary-reward case, complementing the empirical findings.

The KL-forgetting relationship, while correlational rather than fully causal, is practically useful and intellectually interesting. The finding that on-policy training implicitly performs KL minimization has implications beyond catastrophic forgetting.

### Critical Issues Requiring Minor Revisions

Despite the strong contribution, the manuscript has presentation problems that led to reviewer confusion. These are straightforward to fix and do not require additional experiments:

### 1. Implementation Details Must Be Explicit

The most critical issue: the paper doesn't make it sufficiently clear that the RL implementation uses *no KL regularization* and *no clipping*. This is mentioned briefly on line 185 but deserves prominent placement. Reviewer 8HAS's concerns about GRPO-specific components stem directly from this lack of clarity.

The authors clarify in their rebuttal that with single gradient steps per data generation, $\pi_{\text{old}} \approx \pi_{\text{new}}$, reducing the method to standard REINFORCE with normalized rewards. The 1-0 REINFORCE experiment (Figure 4) removes even reward normalization, isolating the on-policy effect. This is essential context that readers need upfront.

*Required fix:* Add a methods box or clear statement in Section 3.1 explicitly stating: ``Our RL implementation uses GRPO with single gradient steps (effectively REINFORCE with normalized rewards), no KL penalty, and no clipping. All observed effects stem from on-policy sampling alone.'' Move this from line 185 to a prominent position.

### 2. Figure 2 Readability

Even with magnification, it's difficult to read. This is the paper's central empirical result and deserves better treatment.

*Required fix:* Enlarge Figure 2 substantially by removing gray spaces.

### 3. KL Measurement Choice Needs Upfront Justification

Several reviewers questioned why KL is measured on new-task data rather than old-task data. The authors' pragmatic justification---that foundation models undergo many training stages making old-task distributions ill-defined or inaccessible---makes sense but appears only in the rebuttal.

*Required fix:* Add this motivation explicitly at the start of Section 4. Include the ParityMNIST correlation plot (showing that new-task KL correlates with old-task KL even with disjoint supports) in the appendix. Discuss the connection to Learning without Forgetting (Li \& Hoiem, 2016) more thoroughly---that work also uses new-task data for computational reasons.

### 4. Theory-Practice Gap Should Be Acknowledged

The theoretical analysis (Theorem 5.2) applies to convex model classes. Reviewer 8HAS correctly notes that neural networks introduce complications. The authors' response (that overparameterization results suggest realizability often holds) is reasonable but this limitation needs explicit discussion in the main text.

*Required fix:* Add a brief paragraph in Section 5.2 acknowledging, for example: ``Our theoretical analysis uses a simplified convex setting to isolate the on-policy effect. Extending our analysis to fully characterize deep network dynamics remains important future work.''

### 5. Additional Minor Revisions

- Include optimizer ablations: The authors ran experiments with RMSProp and AdaGrad showing conclusions hold across optimizers. Add these to the appendix.

- Include IPO baseline: The rebuttal includes IPO results showing consistency with theory. Add these to the paper in the appendix.

- Highlight Figure 6: The finding that SFT + KL regularization has minimal effect while RL + KL regularization helps substantially is an important negative result that should be more prominent.

## Addressing Reviewer Disagreement

The substantial gap between ratings (8, 8, 6, 2) stems primarily from presentation issues rather than fundamental flaws.

**Reviewer 8HAS's concerns (score: 2):** This reviewer raised legitimate questions about GRPO components, optimizer dependence, and theoretical treatment. However, their concerns appear to stem from misunderstandings that the authors addressed convincingly in rebuttal:

- On GRPO components: The reviewer worried that clipping or KL regularization might explain the results. The authors clarify that neither is used. This is definitive but was insufficiently clear in the original manuscript.

- On theoretical treatment: The reviewer asked why SFT doesn't receive analogous theoretical guarantees. The authors' response is correct: SFT's behavior depends entirely on the arbitrary annotator distribution, making distribution-agnostic guarantees impossible. RL's self-generated distribution enables general characterization. This asymmetry is fundamental, not a gap in the analysis.

- On KL measurement: The reviewer argued that KL should be measured on old-task data. The authors' pragmatic justification (infeasibility for foundation models) is sound, and the ParityMNIST experiment validates that new-task KL correlates with old-task KL even with disjoint supports.

- On optimizer dependence: The authors conducted additional ablations with RMSProp and AdaGrad, confirming that conclusions hold across optimizers.

These are all legitimate questions that reveal unclear exposition, but the authors' answers are scientifically sound. With the minor revisions I've outlined, I expect these concerns would be fully resolved.

**Reviewer xH9f's concerns (score: 6):** This reviewer raised valid points about correlation vs. causation and visual presentation. The authors acknowledge that the connection is correlational rather than fully causal---this is stated explicitly in the paper as a limitation. Regarding figure compression, the authors explain that limited compute prevented training to convergence, and offer alternative visualizations. These are reasonable responses that don't undermine the contribution.

**Reviewers Qf7W and muvc (scores: 8, 8):** Both reviewers recognized the novelty and significance of the work, praised the comprehensive evaluation, and found the presentation clear. Their positive assessments align with my own evaluation.

## Why Accept?

This paper makes three concrete contributions:

- Empirical discovery: RL fine-tuning forgets less than SFT at matched new-task performance across multiple domains.

- Empirical forgetting law: KL divergence on new-task data predicts catastrophic forgetting with high fidelity ($R^2 = 0.96$ in ParityMNIST, $R^2 = 0.71$ in LLMs).

- Mechanistic explanation: On-policy RL implicitly minimizes KL divergence through its self-generated training distribution, supported by both theory (Theorem 5.2) and experiments (oracle SFT, ablations).

The work is scientifically sound, practically relevant, and opens concrete research directions. The revisions needed are primarily about clarity and completeness---making explicit what is currently buried or implicit. None require new experiments or substantial additional work.

The disagreement among reviewers reflects presentation issues, not fundamental methodological problems. The authors have demonstrated good faith engagement and provided convincing responses to all substantive concerns.

## Limitations and Future Work

The authors appropriately acknowledge several limitations:

- The KL-forgetting relationship is correlational rather than fully causal. A mechanistic account of why larger KL shifts disrupt prior knowledge remains an open question.

- Theoretical analysis is restricted to convex settings; extending to neural networks requires additional machinery.

- LLM experiments did not train to convergence due to compute constraints and GRPO algorithm nature.

- Only one robotic task is evaluated; off-policy RL methods are not studied.

- Behavior at frontier scales remains unknown.

These are honest limitations that do not diminish the contribution. The paper makes meaningful progress on an important problem and provides actionable insights for practitioners.

## Final Recommendation

With these changes, this will be a strong contribution to ICLR. The work represents an important step toward understanding catastrophic forgetting in foundation models and has clear practical implications for post-training methodology.

**Reviewer Concerns:**

see "Summary" text.

**Reviewer Scores:**

see "Summary" text.

---

### Decision · Program_Chairs · 2026-01-26

Accept (Poster)